# Analysis of foothold selection during locomotion using terrain reconstruction

**Karl S Muller¹†, Kathryn Bonnen²†, Stephanie M Shields², Daniel P Panfili¹, Jonathan Matthis³, Mary M Hayhoe¹\***

¹Center for Perceptual Systems, The University of Texas at Austin, Austin, United States; ²School of Optometry, Indiana University, Bloomington, United States; ³Department of Biology, Northeastern University, Boston, United States

## eLife assessment

This **fundamental** study has the potential to substantially advance our understanding of human locomotion in complex real-world settings and opens up new approaches to studying (visually guided) behavior in natural settings outside the lab. The evidence supporting the conclusions is overall **compelling**. Whereas detailed analyses represent multiple ways to visualize and quantify the rich and complex natural behavior, some of the specific conclusions remain more suggestive at this point. The work will be of interest to neuroscientists, kinesiologists, computer scientists, and engineers working on human locomotion.

**\*For correspondence:**
hayhoe@utexas.edu

†These authors contributed equally to this work

**Competing interest:** The authors declare that no competing interests exist.

**Abstract** Relatively little is known about the way vision is used to guide locomotion in the natural world. What visual features are used to choose paths in natural complex terrain? To answer this question, we measured eye and body movements while participants walked in natural outdoor environments. We incorporated measurements of the three-dimensional (3D) terrain structure into our analyses and reconstructed the terrain along the walker's path, applying photogrammetry techniques to the eye tracker's scene camera videos. Combining these reconstructions with the walker's body movements, we demonstrate that walkers take terrain structure into account when selecting paths through an environment. We find that they change direction to avoid taking steeper steps that involve large height changes, instead of choosing more circuitous, relatively flat paths. Our data suggest walkers plan the location of individual footholds and plan ahead to select flatter paths. These results provide evidence that locomotor behavior in natural environments is controlled by decision mechanisms that account for multiple factors, including sensory and motor information, costs, and path planning.

## Introduction

Sensory input guides actions, and in turn, those actions shape the sensory input. Consequently, to develop a sophisticated scientific understanding of even simple actions in the natural world, we must monitor both the sensory input and the actions. While technology for monitoring gaze and body position during natural behavior is both readily available and widely used in vision science and in movement science, the use of technology for the measurement of the visual input in natural environments has been limited. In this paper, we aim to bridge that gap by using photogrammetry techniques from computer vision to reconstruct the environment and subsequently approximate the visual input. The combination of the reconstructions with body pose information and gaze data allows a full specification of how walkers interact with complex real-world environments. These data can help expand our

understanding of how visual information about the structure of the environment drives locomotion via sensorimotor decision-making.

Natural visually guided behaviors, like visually guided walking, can be characterized as a sequence of complex sensorimotor decisions (*Hayhoe, 2017*; *Gallivan et al., 2018*; *Domínguez-Zamora and Marigold, 2021*). However, much of our current understanding of locomotion comes from work characterizing steady-state walking in laboratory settings – most commonly with participants walking on treadmills. That work has shown that humans converge toward energetic optima. For example, walkers adopt a preferred gait that constitutes an energetic minimum given their own biomechanics (*Warren, 1984*; *Warren et al., 1986*; *Kuo et al., 2005*; *Selinger et al., 2015*; *Finley et al., 2013*; *Lee and Harris, 2018*; *Rock et al., 2018*; *Yokoyama et al., 2018*; *O'Connor et al., 2012*).

There are a number of problems in generalizing these findings to walking in natural environments. In particular, locomotion over rough terrain depends on both the biomechanics of the walker and visual information about the structure of the environment. When the terrain is more complex, walkers use visual information to find stable footholds (*Matthis et al., 2018*). There are also other factors to consider, such as the need to reach a goal or attend to the navigational context (*Warren et al., 2001*; *Rio et al., 2014*; *Logan et al., 2010*; *Patla and Vickers, 1997*). Thus, the sensorimotor decisions in natural locomotion will be shaped by more complex cost functions than in treadmill walking. Furthermore, in the face of this complexity, individuals may be adopting heuristics rather than converging upon optimal solutions.

Previous studies tracking the eyes during outdoor walking have found that gaze patterns change with the demands of the terrain (*Pelz and Rothkopf, 2007*; *Foulsham et al., 2011*; 't *'t Hart and Einhäuser, 2012*). However, in those studies, foot placement was not measured, making it impossible to analyze the relationship between gaze and foot placement. Recent work by *Matthis et al., 2018*, and *Bonnen et al., 2021*, integrated gaze and body measurements of walking in outdoor environments. Those papers demonstrated that walkers modulate gait speed in order to gather visual information necessary for the selection of stable footholds as the terrain became more irregular. Walkers spent more time looking at the ground close to their body (2–3 steps ahead) with increasing terrain complexity. While gaze and gait were tightly linked, the absence of terrain measurements made it impossible to ask what visual terrain features walkers use to choose footholds and navigate toward the goal.

In this paper, we ask how vision is used identify viable footholds and choose paths in natural environments. In particular, what are the visual features of the terrain that underlie path choice? How do walkers use visual information to alter the preferred gait cycle appropriately for the upcoming path? We accomplish this by reconstructing the terrain and aligning the gaze and gait data to that reconstruction. Then, we perform a series of analyses of walkers' body movements and the terrain, demonstrating that: (a) depth information available to walkers is predictive of upcoming footholds, (b) walkers prefer flatter paths, and (c) walkers choose indirect routes to avoid height changes. These findings shed light on how walkers use visual information to find stable footholds and choose paths, a crucial everyday function of the visual system.

## Results

We analyzed data recorded while participants walked over rough terrain (*n*=9). The data were collected by the authors for two separate, previously published studies of visually guided walking (*Bonnen et al., 2021*, *n*=7; *Matthis et al., 2022*, *n*=2). Walkers' eye and body movements were recorded using a Pupil Labs Core mobile binocular eye tracker and a Motion Shadow full-body motion capture suit. Additionally, the walker's view of the scene was recorded by the eye tracker's outward-facing scene camera. As the scene camera moves with the head, the camera's view of the terrain changes along with the walker's. Due to those changes in viewpoint, the scene videos contain information about the terrain's depth structure, which we aimed to recover via photogrammetry.

### Terrain reconstruction

To reconstruct the three-dimensional (3D) environment from the scene videos recorded by the eye tracker's scene camera, we used the photogrammetry software package Meshroom (*Griwodz et al., 2021*), which combines multiple image processing and computer vision algorithms. The terrain

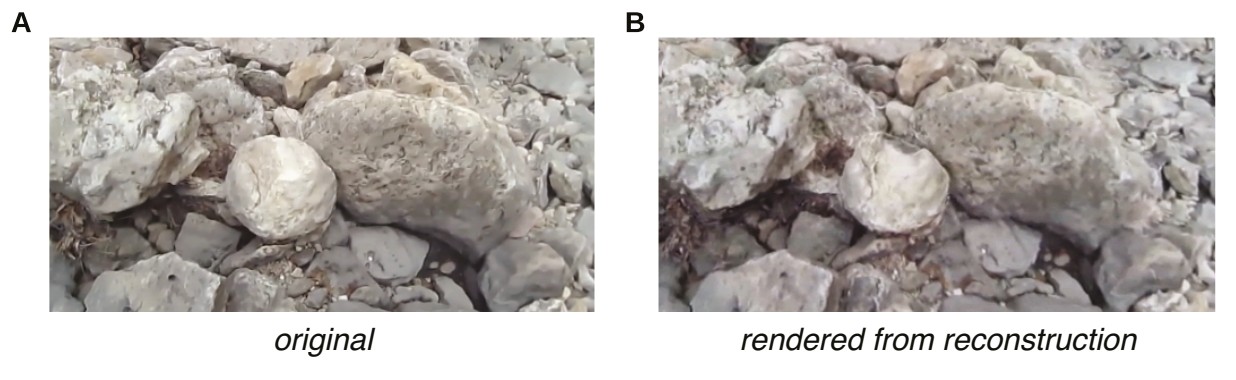

**Figure 1.** Example comparison of original and rendered video frames. We used the scene videos recorded by the eye tracker's outward-facing camera to estimate the structure of the environment and the scene camera's pose in each frame of the video. By moving a virtual camera to those poses and rendering the camera's view of the textured mesh, we can generate comparison images to help assess the reconstruction's accuracy. (**A**) Frame from original scene video. (**B**) Corresponding rendered image.

reconstruction procedure uses the viewpoint changes across each scene video's frames to recover the environment's depth structure. The outputs, generated per scene video, are: (a) a 3D textured mesh of the terrain and (b) an estimate of the 6D camera pose (both 3D location and 3D orientation) within the terrain's coordinate system. To give a sense of the quality of these reconstructions, *Figure 1* shows an example comparison of (A) the original scene camera image and (B) a corresponding view of the reconstructed terrain.

## Aligned gaze, body, and terrain data

We aimed to analyze the body and gaze data in the context of the reconstructed environment, which meant that we needed to align our data on the walker's movements to the terrain's coordinate system. We determined how to position the eye and body movement data relative to the terrain by aligning the head pose measured by the motion capture system to the estimated scene

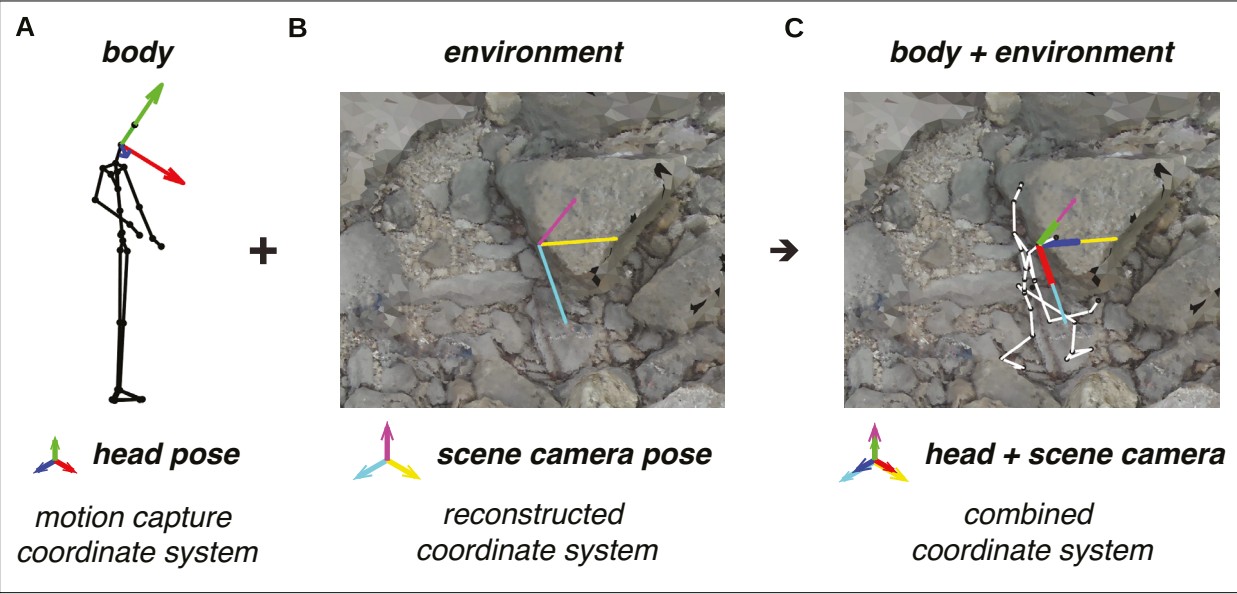

**Figure 2.** Alignment of motion capture data and terrain reconstruction. We combine the motion capture data with the reconstructed environment (photogrammetry data) by aligning the head's pose (RGB axes) to the scene camera's pose (CMY axes). (**A**) Motion capture data provides body pose (i.e. position and orientation) information, including the head's pose (RGB axes). (**B**) The process of reconstructing the environment via photogrammetry produces a three-dimensional (3D) terrain mesh (image) and scene camera's poses (CMY axes). (**C**) Aligning the head's pose (RGB axes) to the scene camera's pose (CMY axes) places the body within the terrain reconstruction.

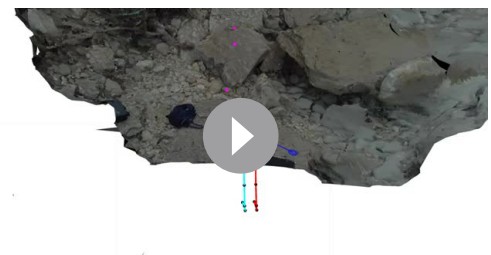

**Video 1.** Visualization of the aligned motion capture, eye tracking, and terrain data for one traversal of the Austin trail: https://youtu.be/TzrA_iEtj1s. The video shows the three-dimensional (3D) motion capture skeleton walking over the textured mesh. Gaze vectors are illustrated as blue lines. On the terrain surface, the heatmap shows gaze density, and the magenta dots represent foothold locations.

https://elifesciences.org/articles/91243/figures#video1

camera pose (*Figure 2*). To visualize the fully aligned data, we created videos showing the walker's skeleton moving through the associated textured terrain mesh (for an example, see *Video 1*).

We also visualized the different paths that walkers took through the terrain. *Figure 3* shows an overhead view of the reconstructed terrains from the Austin dataset, with the paths chosen by the two Austin subjects overlaid onto the terrain. (For examples from the Berkeley dataset, see *Figure 3—figure supplement 1*). The recorded paths were certainly not identical, indicating that foothold locations were not highly constrained. However, the two subjects' paths show considerable regularities. Visual inspection of the paths, particularly in 3D, gives the impression that the terrain's structure impacts the regularity of paths. In other words, features of the 3D environment might impact the degree of variability between paths, suggesting that there may be some identifiable visual features that underlie path choices.

To further illustrate the information present in our dataset, *Figure 4* shows an excerpt of the terrain from *Figure 3* with the following aligned data: gaze locations (green and blue dots), foothold locations (pink dots), and head locations (orange dots).

The close relationship between the gaze locations near the path (green dots) and the foothold locations was the concern of the investigations that generated our dataset: *Matthis et al., 2018*, and *Bonnen et al., 2021*. Those studies showed that gaze was clustered most densely in the region 2–3 steps ahead of the walker's current foothold and ranged between 1 and 5 steps ahead. In other words, we previously found that walkers look close to the locations where the feet will be placed, up to five steps ahead of their current location.

Relevant to the work in this article are the gaze locations 'off-the-path' (blue dots) and the concurrent head locations (connected by blue lines). Those gaze points are off of the walker's chosen path but are still on the ground. Further, they seem to precede turns – a pattern which we observed throughout the data. In later sections, we provide evidence that walkers make a trade-off between maintaining a straight path vs. maintaining a flat path; this gaze pattern points to how the visual system might collect the information used to make that trade-off.

## Benefits of reconstructing terrain

Incorporating the reconstructed terrain into our analyses has several advantages. Perhaps the most obvious is that having information about the terrain's depth structure allows us to analyze the relationship between that depth structure and the walkers' chosen foothold locations. Another major advantage of incorporating the terrain reconstruction is that it enables more accurate gaze and body localization. In previous work, we assumed a flat ground plane, which led to parallax error in gaze location estimates (*Matthis et al., 2018*). Here, we used the reconstructed 3D terrain surface, eliminating the need to assume a flat ground plane and thus eliminating that source of error. Additionally, body position estimates were previously negatively impacted by inertial measurement unit (IMU) drift, which results from the accumulation of small errors in the accelerometer and gyroscope measurements over time. This drift causes global, i.e., not local error – error in the overall localization of the motion capture skeleton, not in the localization of different body parts relative to one another. We were able to address this source of error by fixing the body's reference frame to that of the environment (*Figure 2*). Thus, by eliminating both of these sources of error, utilizing photogrammetry allowed us to more precisely estimate gaze and foothold locations.

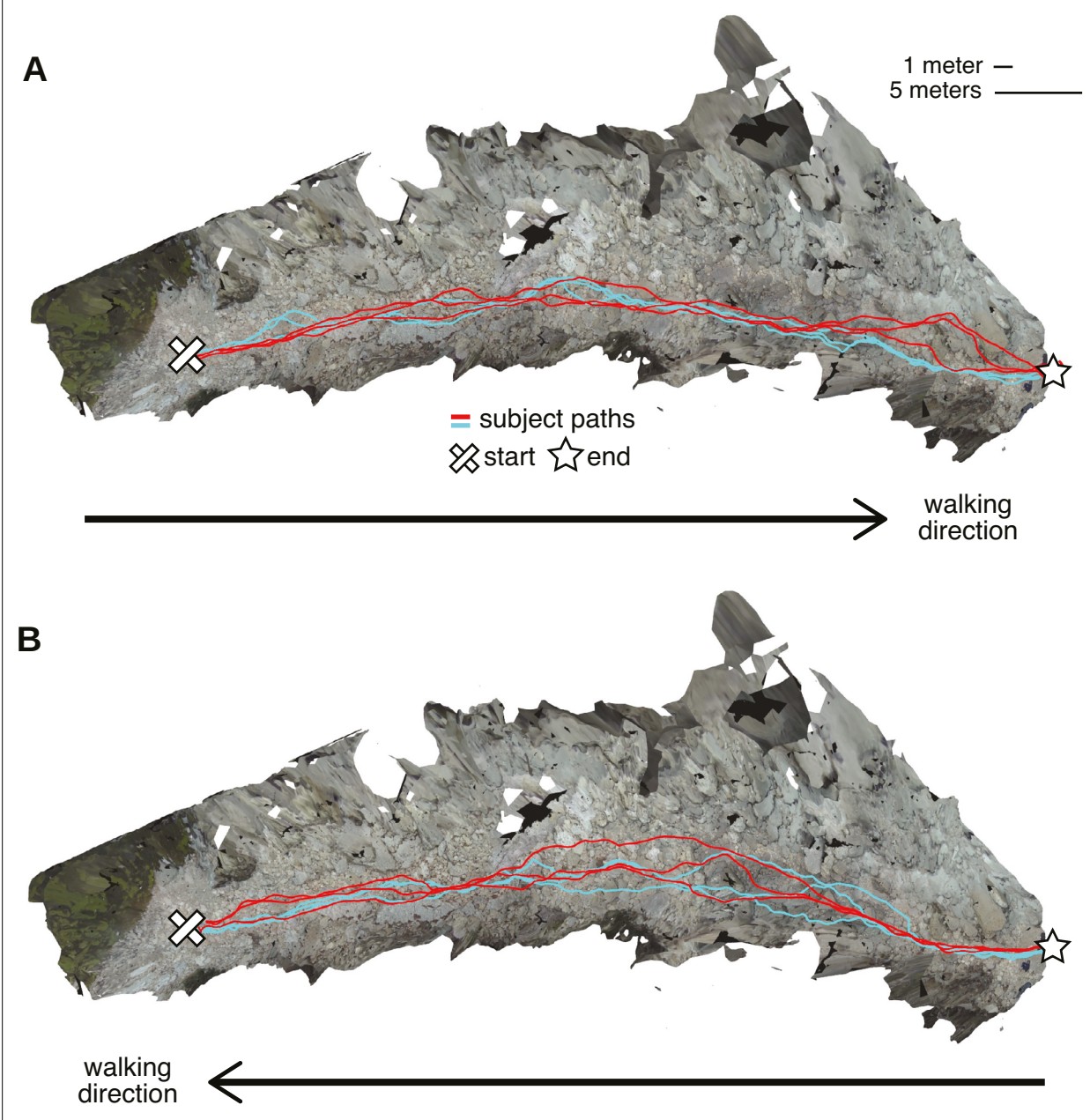

**Figure 3.** Repeated walks across rough terrain in the Austin data. Each of the two Austin subjects walked out and back over a rocky trail three times. These overhead views show the textured three-dimensional (3D) terrain mesh along with the paths the subjects took through that terrain. Each color (red and cyan) corresponds to a different subject. Note that in some sections of the terrain, paths were highly similar across repetitions and across subjects, while in other sections, paths differed notably. (**A**) Both subjects' three walks from the start of the path to the end. (**B**) Both subjects' three walks returning to the start location.

The online version of this article includes the following figure supplement(s) for figure 3:

**Figure supplement 1.** Berkeley path consistency, convergence and divergence.

### Evaluating reliability of terrain reconstruction

To evaluate the reliability of the 3D reconstruction procedure, we compared the terrain meshes calculated from multiple traversals of the same terrain. We used the Austin dataset for this reliability analysis because the terrain is contiguous, the walking paths have clearer start/end points, and there are 12 traversals of the terrain (2 subjects, 6 traversals each; see *Figure 3*).

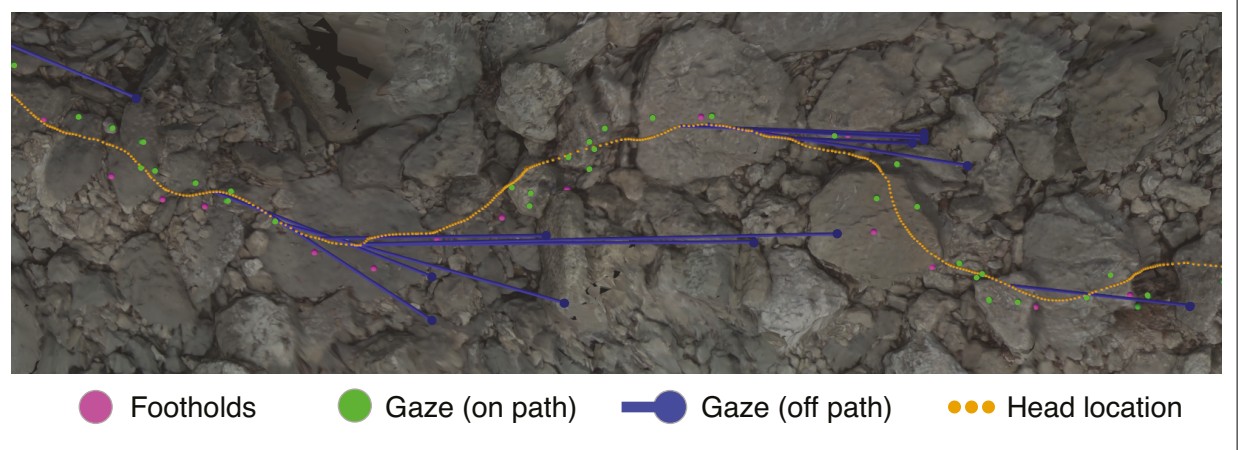

**Figure 4.** Gaze and body data embedded in the corresponding reconstructed terrain. This overhead view shows a representative excerpt of 20 steps from one of the Austin traversals. The walker was, in this overhead view, moving left to right. Dots mark the footholds locations (pink), gaze locations near the path (green), gaze locations off the path (blue), and head locations (orange). To illustrate the relationship between 'off-the-path' gaze and head location, blue lines connect each blue gaze point to the simultaneous location of the head.

Since we performed the reconstruction procedure on each traversal separately, we generated 12 Austin meshes that represented the same physical terrain. Each mesh contained a cloud of points, which we aligned and compared using CloudCompare (https://www.danielgm.net/cc/). For each pair of meshes, one mesh served as the baseline mesh, and one mesh served as the comparison mesh. For each point in the baseline mesh that was within 2 m of the walking path, we found its nearest neighbor in the comparison mesh and calculated the distance, resulting in a distribution of distances (errors) between the two meshes.

The aggregate distribution across all pairwise mesh comparisons is shown in *Figure 5A*. The median error was 4.5 cm, and the 95% quantile was 20.0 cm. To evaluate reliability specifically at footholds, we also isolated the points in each baseline mesh associated with foothold locations (*Figure 5B*). For the foothold locations, the median error was 4.0 cm, and the 95% quantile was 16.8 cm. To put these numbers into context, the average foot length of a person from North America is 25.8 cm (*Jurca et al., 2019*), so in both cases, the majority of mesh errors fall below 20% of the average foot length.

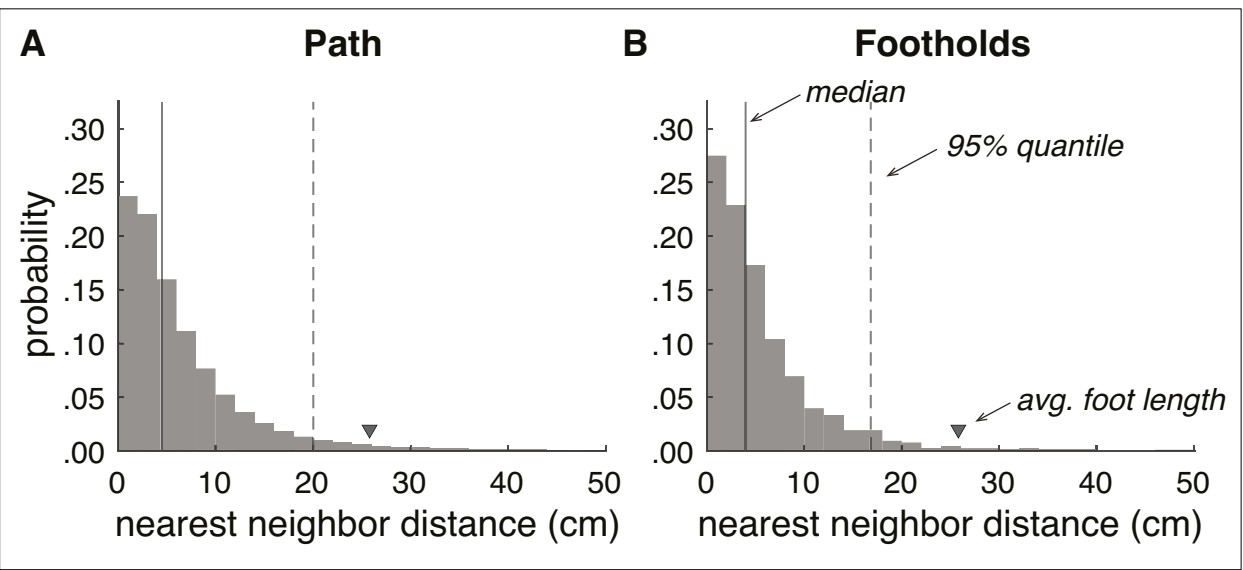

**Figure 5.** Accuracy of terrain reconstructions. (**A**) Nearest neighbor error distribution for the whole terrain (median = 4.5 cm, 95% quantile = 20.0 cm). (**B**) Nearest neighbor error distribution for individual footholds (median = 4.0 cm, 95% quantile = 16.8 cm).

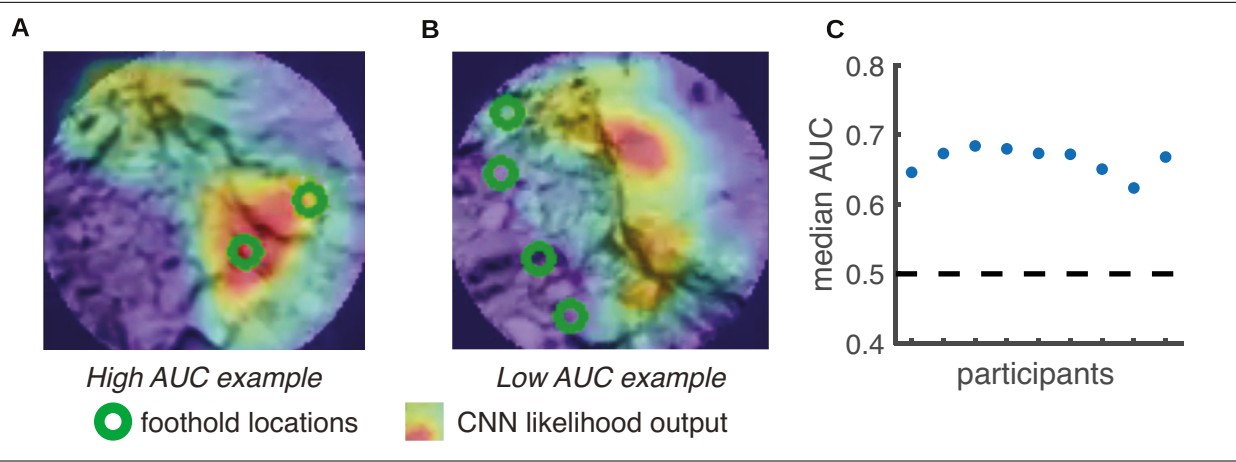

**Figure 6.** Predicting foothold locations from depth information. A convolutional neural network (CNN) was trained to predict foothold locations in retinocentric depth images. (**A**) Example retinocentric depth image associated with relatively good CNN performance (i.e. a high area under the ROC curve [AUC] value). The image is overlaid with the foothold locations (green) and a heatmap showing the CNN's likelihood output, which indicates the likelihood of finding a foothold in a particular location. (**B**) Example retinocentric depth image associated with relatively poor CNN performance (i.e. a low AUC value). (**C**) Median AUC values for the data of each of the nine participants.

Thus, our terrain reconstruction procedure produces reasonably reliable reconstructions of a walker's 3D environment.

## Retinocentric depth information affects foothold selection

The results of *Bonnen et al., 2021*, suggest depth judgments are important in foothold finding, as the removal of depth information shifts gaze to foothold locations that are closer to the walker. Taking advantage of the output of the terrain reconstruction procedure, we sought to confirm that depth information from the walker's point of view could be used to predict the upcoming foothold locations.

We first used the reconstructed terrain – along with the aligned foothold and gaze information – to generate retinocentric depth images that approximate the visual information subjects have access to during walking (e.g. see Figure 11). Note that for each frame in the training dataset, the camera field of view includes multiple future footholds (up to 5; depicted as green circles in *Figure 6A and B*). We used the location of the footholds in these retinocentric depth images to create a training dataset. If a convolutional neural network (CNN) can predict foothold locations above chance based on these retinocentric depth images, that would suggest that terrain depth structure plays a role in foothold selection.

Per subject, we trained the network on half of the terrain and tested on the remaining half (ensuring that the network was tested on terrain that it had not previously seen). For each depth image, we calculated the area under the ROC curve (AUC), which quantifies the discriminability between image locations that show footholds and image locations that do not show footholds. *Figure 6C* shows that, per subject, the median AUC value for depth images from the test set was above chance. We can thus conclude that the network was able to find the potential footholds in the depth images, suggesting that retinocentric depth information contributes to foothold finding.

## Walkers prefer flatter paths

Our CNN analysis suggested that depth features in the upcoming terrain have some predictive value in the selection of footholds. We next decided to narrow our focus to examine whether terrain height, specifically, might play a role in foothold selection. Stepping up and down is energetically costly, and previous work in simpler environments has demonstrated that humans attempt to minimize energy expenditure during locomotion (*Selinger et al., 2015*; *Finley et al., 2013*; *Lee and Harris, 2018*). Furthermore, a walker avoiding large steps up and down would be choosing to take a flatter path, and walking on a flatter path would result in less deviation from their preferred gait cycle and thus more stable locomotion. To test the hypothesis that walkers seek out a flatter path to avoid large changes

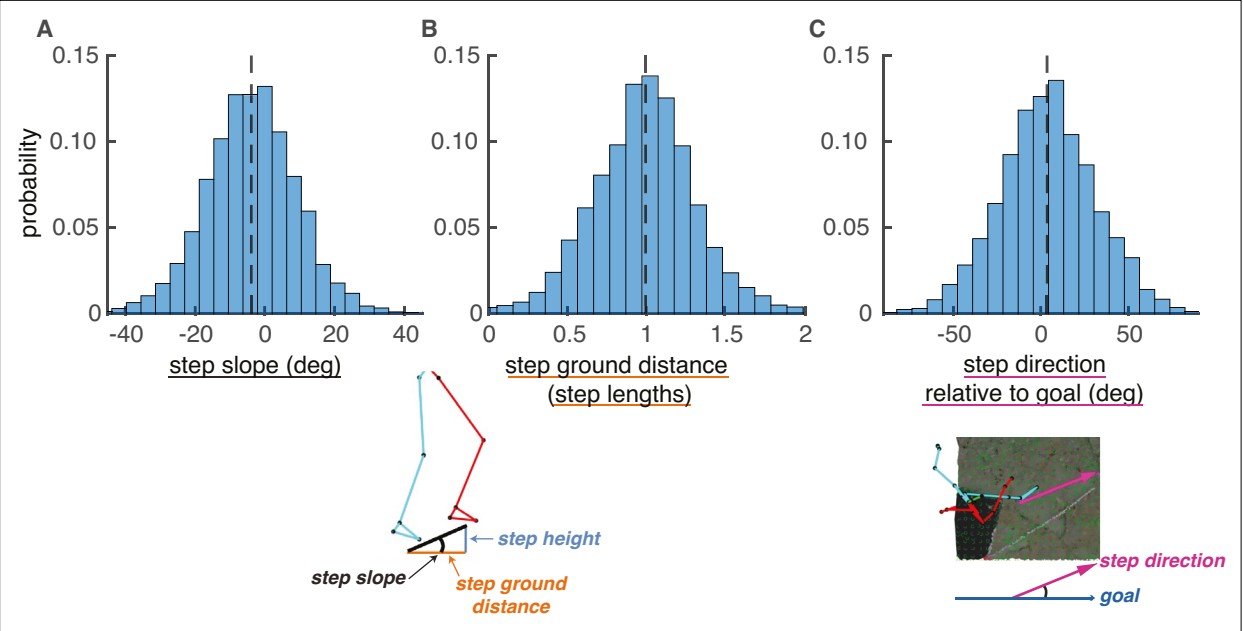

**Figure 7.** Step parameter distributions help define feasible alternative paths. The histograms show the distributions of (**A**) step slopes, (**B**) step lengths, and (**C**) step direction relative to goal direction. These distributions define the set of feasible next steps for a given foothold, allowing the calculation of feasible alternative paths to the one actually chosen by the subject. This figure shows histograms of these quantities pooled over subjects, but note that calculations of viable paths were done based on the step parameters of each individual subject.

The online version of this article includes the following figure supplement(s) for figure 7:

**Figure supplement 1.** Per-subject histograms of the step parameters shown in *Figure 7*.

**Figure supplement 2.** Aggregate histograms of the step parameters defined in the Methods section titled 'Step analysis'.

**Figure supplement 3.** Scatterplots between all parameters defined in the Methods section titled 'Step analysis'.

in terrain height, we measured the slope of steps chosen by our walkers and compared them to the slope of steps in paths we simulated along the same terrain.

We simulated plausible paths for each walker that were comparable to their chosen paths. To ensure plausibility, we constrained potential foothold locations and potential steps based on human walking behavior. We identified potential foothold locations based on the maximum walk-on-able surface slant reported in *Kinsella-Shaw et al., 1992*, excluding areas of the terrain with an average local surface slant greater than 33°. We identified potential steps between foothold locations based on each walker's data, excluding steps with a step slope (*Equation 7*), step ground distance (*Equation 2*), and/or step deviation from goal (*Equation 5*) greater than the corresponding maximum absolute value for that walker's chosen steps (*Figure 7*). Only foothold locations and steps that met these conditions were considered viable.

To ensure comparability, we split each walker's path into 5-step (i.e. 6-foothold) segments, and we simulated corresponding 5-step sequences by starting at the walker's chosen foothold and randomly choosing each subsequent step from the available viable options (*Figure 8A*). (Note that, for this simulation, we did not predefine the end points of path segments).

After simulating walkable path segments that we could directly compare with walkers' chosen path segments, we calculated the overall slope of each path segment by averaging the absolute slope of the steps in the sequence. The resulting path segment slope values quantify the net flatness of each path. The per-subject median chosen path segment slope ranged from 7.7° to 11.8°, with a mean of 9.5° and a standard deviation of 1.7°. This corresponds to quite a small change in height; for a step of average length, a 9.5° slope corresponds to a height change of just a few inches. The per-subject median simulated path segment slope ranged from 9.1° to 19.2°, with a mean of 14.9° and a standard deviation of 3.3°.

As is evident from the per-subject medians, the slopes of chosen path segments tended to be lower than the slopes of simulated path segments, consistent with the idea that walkers seek to

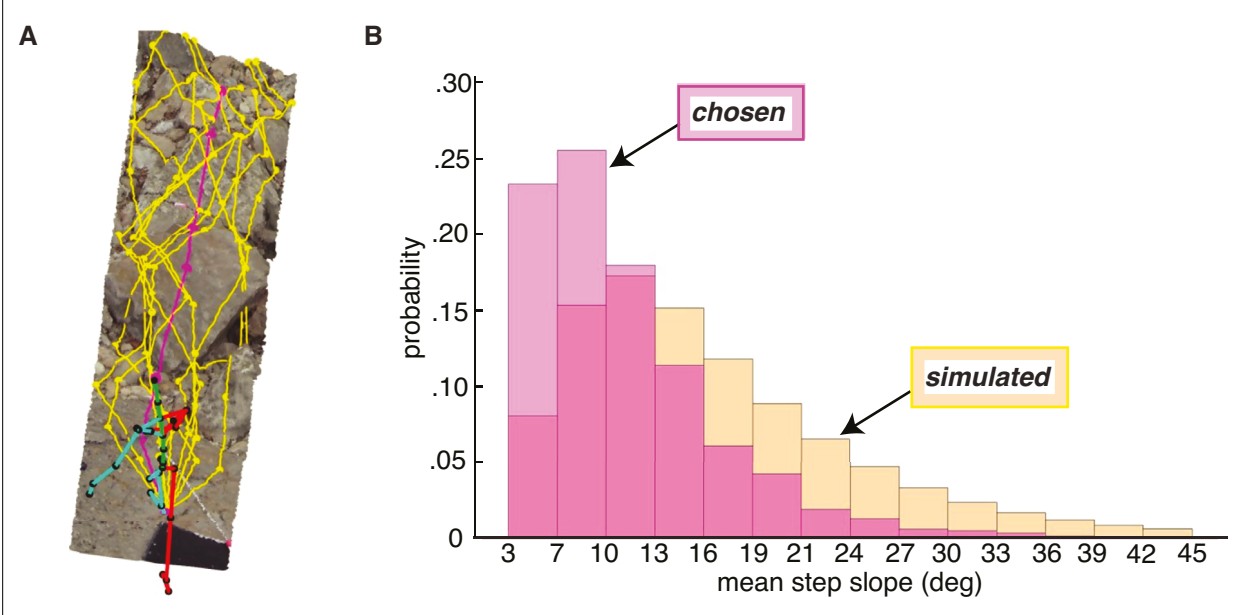

**Figure 8.** Paths chosen by walkers have a lower step slope. We simulated path segments composed of viable steps and compared them to subjects' chosen step sequences. (**A**) Overhead view of an example chosen step sequence (magenta), along with a subset of the corresponding simulated viable step sequences (yellow). The cyan and red lines show the walker's skeleton. (**B**) Histograms of mean step slope for chosen and simulated step sequences for one participant.

minimize energetic costs by taking flatter routes. The bias in the chosen path segment slope distribution toward lower slopes (vs. the simulated distribution) can be seen in *Figure 8*, where both path segment slope distributions for one example subject are plotted. For every subject, their median chosen path segment slope was lower than their median simulated path slope, with the simulated slopes being 5.56° larger on average (SD = 2.18°). A paired sample *t*-test confirmed that the differences between the two medians were statistically significant, $t(8) = 7.64$, $p \ll 0.001$.

Our results suggest that walkers prefer taking flatter paths. This does not mean that they categorically avoid large steps up and down. Clearly they do sometimes choose paths with greater slopes, as is indicated by the tail of the chosen distribution. However, on average they tend toward flatter paths than would be predicted by the terrain alone.

## Walkers choose indirect routes to avoid height changes

Another factor that influences the energetic cost of taking a particular path is how straight the path is. Changing direction requires more energy than walking on an equivalent straight path (*McNarry et al., 2017*), as one might expect since curvier (i.e. more tortuous) paths are longer and require walkers to deviate from their preferred gait cycle. However, walkers frequently alter their direction of travel in rocky terrain (see, e.g., *Figure 4*). If there is a large height change along the straight path, turning might require less energy than stepping up or down while following the straight path. Therefore, building on our finding that walkers prefer flatter paths, we hypothesized that walkers choose to turn when turning allows them to avoid notable changes in terrain height. We evaluated that possibility by examining the relationship between the tortuosity of their chosen path segments and the slope of corresponding straight alternative path segments.

As in the prior section, we simulated path segments to compare walkers' chosen steps to the viable steps along that specific terrain. We followed a similar procedure with one notable difference: Here, for each chosen path segment, we simulated steps between the first and sixth footholds in the chosen path segment. In other words, we predefined both the starting and ending locations of simulated path segments (*Figure 9A*), whereas above, we predefined only the starting locations (*Figure 8A*).

We used the simulated path segments to quantify, for each chosen path segment, the average step slope a subject would encounter if they tried to take a straighter path between the segment's end points. To accomplish that, we quantified the straightness of all path segments via a tortuosity

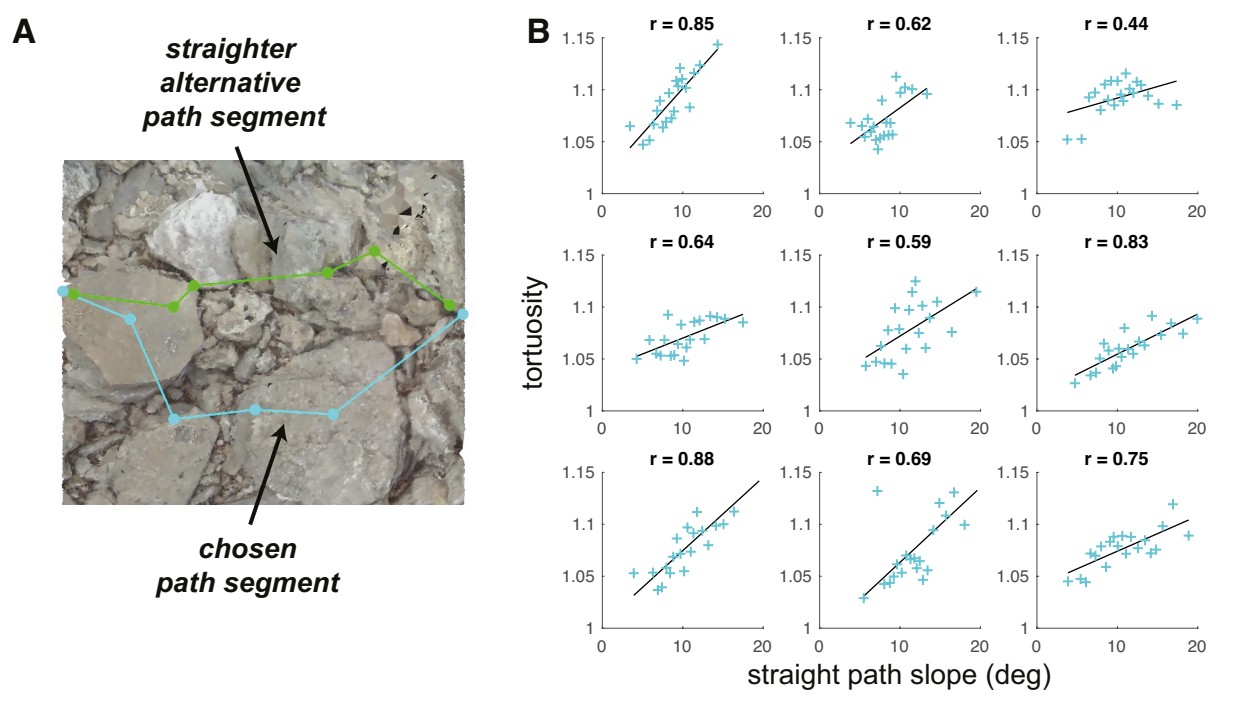

**Figure 9.** Average tortuosity of chosen path increases with increased straight path slope. (**A**) An example chosen path segment (cyan; 5-step sequence), along with one straighter alternative path segment (green). (**B**) An illustration of the relationship between chosen path segment tortuosity and the slope of 'straight' path segments that were simulated across the same terrain. Each subpanel depicts one subject's data. To summarize the large amount of data per subject (317,497 path segments), we binned the data into 20 quantiles of straight path slope and averaged tortuosity per bin, generating one summary tortuosity value per slope level. These scatterplots show the average tortuosity as a function of the straight path slope quantile (cyan crosses), along with best fit lines (black). (For scatterplots showing data per chosen path segment, see *Figure 9—figure supplement 1A*). Associated correlation values (Pearson's *r*) are shown at the top of each panel.

The online version of this article includes the following figure supplement(s) for figure 9:

**Figure supplement 1.** Relationship between average straight path segment slope and the tortuosity of each chosen path segment.

**Figure supplement 2.** Distributions of straight path slope and chosen path segment tortuosity.

metric (*Batschelet, 1981*; *Benhamou, 2004*), and per terrain mesh, we used the tortuosity of chosen path segments to compute a conservative 'straightness' threshold. We then selected the simulated path segments with a tortuosity below the computed threshold, computed their slope, and averaged across those path segment slopes. Those calculations resulted in, for each path segment, the mean slope of relatively straight alternative paths, which we refer to here as 'straight path slope'. *Figure 9A* shows one example straight path segment, together with the path that the subject actually chose.

To determine whether subjects chose longer paths as the slope of relatively straight options increased, we compared path segment tortuosity to the corresponding straight path slope. The tortuosity of chosen path segments ranged from 1 (which denotes a straight path) to 14.73 (which denotes a quite curvy path), with a mean of 1.09 and a standard deviation of 0.29. For all walkers in our dataset, the distribution of tortuosity values was concentrated near 1, with median values per subject of 1.041.07 (M=1.06, SD = 0.01; *Figure 9—figure supplement 2*, left). The simulated straight path slopes ranged from 1.89° to 21.72°, with a mean of 10.31° and a standard deviation of 3.44°. There was some variability in the per-subject distributions, with median values of 7.83°–11.94° (M=10.12°, SD=1.20°; *Figure 9—figure supplement 2*, right), but the primary difference was between Austin and Berkeley participants, suggesting that variability was at least somewhat a function of the terrain.

We expected that walkers would choose curvier (i.e. more tortuous) paths when the relatively straight alternatives were also relatively steep (i.e. have a relatively high slope). Thus, we expected that, when directly comparing the tortuosity and straight path slope values for each chosen path segment, (a) relatively low straight path slope values would be associated with tortuosity values fairly close to 1 and (b) tortuosity values would tend to increase as straight path slope increased. For each

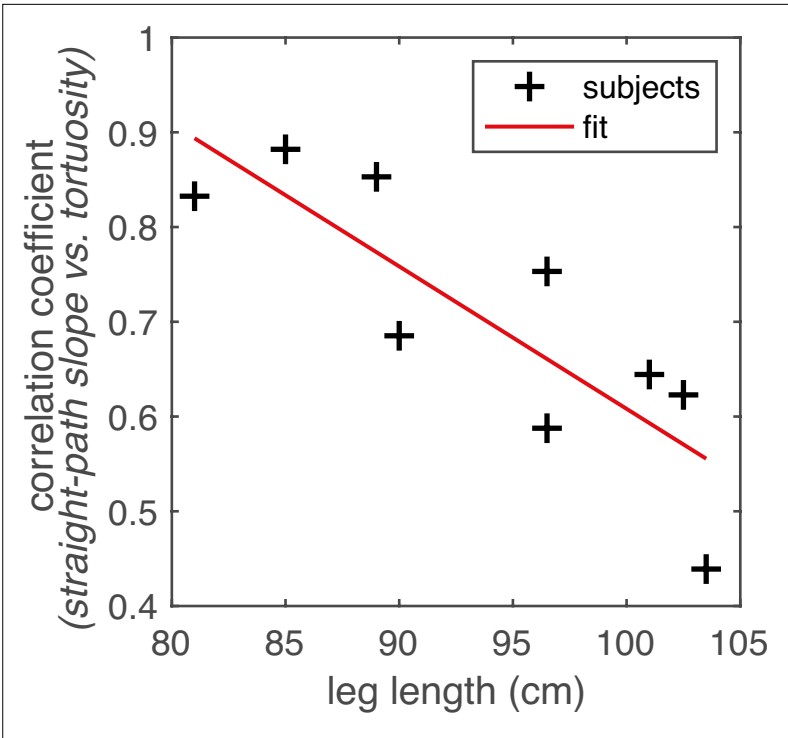

**Figure 10.** Relationship between leg length and the correlation between straight path step slope and path tortuosity. Subjects' leg lengths (in cm) are plotted on the horizontal axis. The correlation coefficients drawn from the analyses depicted in *Figure 9B* are plotted on the vertical axis. The scatterplot shows one point per subject (black crosses). The linear trendline is also shown (red line). We found that there was a statistically significant negative correlation between subjects' leg lengths and the straight path slope vs. average path tortuosity correlations in their data ($r=-0.83$, $p=0.005$). (For a comparable plot showing the correlations derived from data per chosen path segment, see *Figure 9—figure supplement 1B*).

subject, we calculated the average tortuosity of the chosen paths for different levels of 'straight path slope' (*Figure 9B*), and we found that, indeed, (a) average tortuosity values were near 1 at the lowest slope level and average tortuosity increased across increasing quantiles of straight path slope. This relationship suggests that walkers made a trade-off, choosing paths that were flatter but more tortuous over paths that were steeper but more direct. Such choices may reflect decisions that the cost of taking the longer, flatter path is ultimately less than cost of taking the straighter, steeper path. All subjects show this relationship, though its strength does vary between subjects.

The energetic cost of taking steeper steps likely varies with factors affecting a walker's biomechanics. The question of whether a flatter, more torturous path is a more energetically efficient path than a steeper, straighter path likely has a walker-specific answer. Therefore, one might expect that the some of the between-subject variability in the strength of the slope-tortuosity correlation is due to biomechanically relevant factors, such as the walker's leg length. We thus asked whether the strength of the trade-off between tortuosity and straight path slope (i.e. the correlation coefficient) varied with walkers' leg lengths. Subject leg lengths ranged from 81 cm to 103.5 cm, and slope-tortuosity correlation values ranged from 81 cm to 103.5 cm (*Figure 9B*). Longer leg lengths were associated with lower correlation coefficients ($r=-0.83$, $p=0.005$; *Figure 10*), suggesting that subjects with the shortest legs are more likely to choose longer paths when the straight path becomes less flat (i.e. with increasing values of straight path slope).

Note that *Figure 9* and *Figure 10* both use the aforementioned binned data, with the average tortuosity of chosen paths calculated for each of 20 'straight path slope' quantiles to summarize the per-path-segment data. Parallel plots made using the per-path-segment data are shown in *Figure 9—figure supplement 1*. Importantly, the analysis of per-path-segment data reveals similar relationships as those described above. For eight of the nine subjects, the correlations are significant. The correlation coefficients between straight path slope and chosen path tortuosity are substantially smaller for

the per-path-segment data. The lower correlation values result from the amount of variability across path segments. The variability can be seen in the spread of the points shown in *Figure 9—figure supplement 1A* (vs. *Figure 9*), but note that even those subplots do not show the full extent of the variability. Those scatterplots include the full range of straight path slope values (1.89–21.72°) but, in effect, omit the tail of the tortuosity distribution (max = 14.73) to ensure that the majority of the data is readily visible. This cross-path variability suggests that subjects are not using strict criteria to make decisions about the trade-off between path slope and path curvature. Rather, the trends we observe likely reflect learned heuristics about which paths are more or less preferable, which walkers can use to flexibly select paths.

## Discussion

In this work, we present novel analyses of natural terrain navigation that take advantage of the 3D terrain reconstructions we generated using photogrammetry. The terrain reconstructions allowed for greater precision than was possible in previous studies of walking in natural outdoor environments (*Matthis et al., 2018*; *Bonnen et al., 2021*). We were able to more accurately calculate both gaze and foothold locations. Most importantly, the quantification of terrain geometry allowed us to examine how the structure of the visual environment influences foothold selection. An analysis of this relationship – between the structure of the visual environment and selected footholds/paths – has been missing in much previous work on visually guided action in the natural world, where the depth structure is typically not measured.

After developing the reconstruction and data alignment procedure, our next challenge was to develop a strategy for identifying visual features that influenced subjects' foothold and path selection. We noted regularities in the paths chosen by walkers, both across individuals and across repeats of the same walk, suggesting that there are some terrain features that serve as a basis for path choice. Previous work suggested a role for depth features in visually guided walking (*Bonnen et al., 2021*). Using a CNN to predict foothold locations on the basis of retinocentric depth images, we confirmed a role for depth information in foothold selection. This result justified the further exploration of depth variation in the terrain (e.g. changes in terrain height) as a potential feature used by walkers in foothold selection.

To ask whether changes in terrain height (i.e. depth structure) influenced path selection, we simulated viable paths that could be compared with the chosen paths. Comparing the sets of chosen and viable paths, we found that walkers prefer flatter paths and avoid regions with large height irregularities. While in some ways this might not be a surprising result, the data reveal that this is a strong constraint on path choice. The median slope of 5-step paths was less than 10°, which corresponds to a quite small height change of about 14–17 cm.

This work did not investigate which depth cues walkers used to make these path choices, but we highlight that gap in knowledge here as an avenue ripe for future study. A variety of depth cues might be relevant to such sensorimotor decisions, including motion parallax generated by the movement of the head, binocular disparity, local surface slant, and the size of the step-able area. Determining how depth cues are used to make these sensorimotor decisions will require more controlled experiments.

Finally, we observed that walkers chose longer paths when the straightest viable paths involved greater height changes (*Figure 9*), and further, our data suggest that walkers were more likely to choose longer paths if their legs were shorter (*Figure 10*). This suggests that the sensorimotor decision-making that supports walking complex terrain is highly body-specific, taking into account the details of a walker's body, like leg length. This suggests that any cost function or model describing the sensorimotor decision-making processes that support walking in complex natural terrains will also need to be body-specific.

## Cost functions in visually guided walking

While we do not know what contributes to the internal cost functions that determine walkers' choices, the preference for flatter paths is likely driven in part by the energetic cost of stepping up or down. On flat ground, humans converge to an energetic optimum consistent with their passive dynamics (*Kuo et al., 2005*; *Selinger et al., 2015*; *Finley et al., 2013*; *Lee and Harris, 2018*). Deviations from this gait pattern, including turns and changes in speed, are energetically costly (*Voloshina et al., 2013*;

*Soule and Goldman, 1972*). Recent work by *Darici and Kuo, 2023*, also showed that subjects are able to minimize energetic cost on uneven ground planes by modulating speed. Our findings suggest that walkers may be adjusting their behavior to minimize energetic costs in natural outdoor terrains as well. Future work should examine more directly how particular walking decisions impact energetic costs in natural outdoor terrains.

### Path planning

Our analyses show that vision is used to locate flatter paths in upcoming steps. We found that the average step slope of the chosen path was significantly lower than simulated paths, suggesting that walkers were intentionally maintaining a flatter path. Furthermore, our findings suggest that walkers turn to avoid paths with large changes in terrain height. To accomplish this, walkers must plan ahead, demonstrating that planning is an important component of path selection in rugged terrain. Though this study has not explicitly examined the role of gaze in walking, future studies of gaze during walking will be critical to understanding how individuals perform path planning.

Laboratory studies suggest that walkers need to look 2 steps ahead to preserve inverted pendulum dynamics (*Matthis et al., 2017*). Biomechanical models indicate that walkers can adjust their gait to accommodate upcoming obstacles and may plan up to 8 or 9 steps ahead (*Darici and Kuo, 2023*). Our previous work studying gaze suggests that, in rocky terrain, walkers distribute most of their gaze on the ground to footholds up to 5 steps ahead (*Matthis et al., 2018*; *Bonnen et al., 2021*). Because of the differences between these studies, it is difficult to say exactly what causes the discrepancy (5 steps vs. 9 steps) in the planning horizons reported in these two studies. However, there are notable differences between the laboratory obstacle paths they used and our natural environments. Their walking paths involved height changes of no more than 7.5 cm, the surfaces themselves were flat, and the path required no changes in direction. Our terrains involved greater height changes, irregular and sloping surfaces, large boulders, and frequent direction changes based on visual information. More complex terrains may also impose a greater load on visual working memory (*Lin and Lin, 2016*). Thus, a shorter planning horizon in our data might be expected as individuals adjust their planning horizon depending on the nature of the terrain. On the other hand, because there is no eye tracking in *Darici and Kuo, 2023*, we cannot rule out the possibility that these two planning horizons are in fact the same – individuals may be able to get information about 8–9 steps ahead from their peripheral vision. More study is needed on the details of planning horizons in walking and how individuals adjust them depending on the task and terrain.

### Conclusion

In conclusion, we have integrated eye tracking, motion capture, and photogrammetry to create a visuomotor dataset that includes gaze information, body position data, and accurate 3D terrain representations. The reconstructed 3D terrains were a valuable addition to our methodology because they allowed a much more direct, more precise investigation of the visual terrain features that are used to guide path choice. Previous investigations of walking in natural outdoor environments have been limited to video recordings. The reconstruction and integration procedures outlined in this paper should be generally useful for the analysis of visually guided behavior in natural environments. In our analyses, we observed that visual information about depth appeared to play a role in path choice. Despite the complexity of the sensory-motor decisions in natural, complex terrain, we observed that there were consistencies in the paths walkers chose. In particular, walkers chose to take more indirect routes to achieve flatter paths, which required them to plan ahead. Taken together, these findings suggest that walkers' locomotor behavior in complex terrain reflects sensorimotor decision mechanisms that involve different costs, sensory and motor information, and path planning.

## Methods

### Experimental data

The data used here was collected by the authors in two separate studies, conducted in two separate locations: Austin, Texas (*Matthis et al., 2022*) and Berkeley, California (*Bonnen et al., 2021*). The studies were approved by the Institutional Review Boards at the University of Texas at Austin and the University of California, Berkeley, respectively. All participants gave informed consent and were

**Table 1.** Information about participants included in dataset.
Table includes the location of data collection, key demographics, and the amount of data recorded per participant (quantified as the number of steps in rough terrain in participants' processed data).

| Location | TX | TX | CA | CA | CA | CA | CA | CA | CA |
|---|---|---|---|---|---|---|---|---|---|
| Age | 23 | 25 | 27 | 39 | 34 | 29 | 24 | 24 | 54 |
| Gender | F | M | M | M | M | F | F | F | M |
| Leg length (cm) | 89 | 102.5 | 103.5 | 101 | 96.5 | 81 | 85 | 90 | 96.5 |
| Step count | 468 | 347 | 462 | 489 | 385 | 537 | 486 | 603 | 453 |

treated according to the principles set forth in the Declaration of Helsinki of the World Medical Association. Both studies used the same eye and body tracking equipment as well as the same data collection methods. Additionally, both included multiple terrain conditions. One terrain condition common to both was rough terrain, which consisted of large rock obstacles with significant height deviations.

## Data selection

In our combined dataset, we included data from only (a) rough terrain, (b) participants walking with normal or corrected-to-normal vision, and (c) participants with scene videos of sufficiently high quality for terrain reconstruction. We therefore did not include any of the Berkeley data used to study the impact of binocular visual impairments. Further, we excluded one Austin participant and one Berkeley participant because the quality of their scene videos caused issues with the terrain reconstruction process, which was essential for the analyses we describe here. (More specifically, one Austin participant was excluded because the scene camera was angled too far upward, limiting the view of the ground, and one Berkeley participant was excluded because their scene videos were too low contrast due to the dim outdoor lighting conditions at the time of the recording).

## Participants

We used data from 9 participants: 2 from the Austin study and 7 from the Berkeley study (*Table 1*). All had normal or corrected-to-normal vision. There were 5 male and 4 female subjects. They were 23–54 years of age at the time of data collection, with an average age of 31 years (median: 27). Their legs were 81–103.5 cm long, with an average of 93.9 cm (median: 96.5 cm).

The amount of data recorded per participant varied since they were tasked with walking along loosely defined paths, rather than walking for a fixed duration, number of steps, etc. In *Table 1*, we represent the amount of data recorded via the number of steps in rough terrain in each participant's processed data. In total, the dataset included 4230 steps. Per participant, there were 347–603 steps, with an average of 470 steps (median: 468). Overall, participants with longer legs took fewer steps ($r=-0.57$), and the Berkeley participants took approximately 125 steps more than Austin participants with similar leg lengths.

## Equipment

Eye movements were recorded using a Pupil Labs Core mobile eye tracker and the Pupil Capture app (Pupil Labs, Berlin, Germany). The eye tracker had two infrared, eye-facing cameras, which recorded videos of the eyes at 120 Hz with 640×480 pixel resolution. Additionally, there was an outward-facing camera mounted 3 cm above the right eye, which recorded the scene in front of the subject at 30 Hz with 1920×1080 pixel resolution and a 100° diagonal field of view. A pair of dilation glasses was fitted over the eyes and eye-facing cameras to protect the infrared eye cameras from interference due to the sun. For participants, this felt like wearing a pair of sunglasses.

Body movements were recorded using Motion Shadow's full-body motion capture suit and the Shadow app (Motion Shadow, Seattle, WA, USA). The suit included 17 IMUs, which each contained three three-axis sensors: an accelerometer, a gyroscope, and a magnetometer. The Shadow app recorded data from the suit at 100 Hz and simultaneously estimated the joint poses (i.e. positions and orientations) for the full 30-node 3D skeleton. IMUs were placed on the head, chest, and hips as well as on both the left and right shoulders, upper arms, forearms, hands, thighs, calves, and feet. The

3D skeleton then included nodes for the head, head top, neck, chest, body, hips, mid spine, and low spine as well as the left and right shoulders, arms, forearms, hands, fingers, thighs, legs, feet, toes, heels, and foot pads.

In addition to the eye tracker and motion capture suit, subjects wore a backpack-mounted laptop, which was used to record all raw data. Importantly, using the same computer to record both data streams meant both were recording timestamps queried from the same internal clock. Their timestamps were therefore already synchronized.

## Task

At the beginning of each recording, participants performed a 9-point vestibulo-ocular reflex (VOR) calibration task. They were instructed to stand on a calibration mat 1.5 m from a calibration point marked on the mat in front of them. This distance was chosen based on the most frequent gaze distance in front of the body during natural walking in these terrains (*Matthis et al., 2018*). They were instructed to fixate on the calibration point while rotating their head along each of the eight principal winds, i.e., the four cardinal and four ordinal directions. Their resulting VOR eye movements were later used to calibrate the eye tracking data and to spatially align the eye and body data.

Participants' primary task was to walk along a trail. The Austin participants walked along a rocky, dried out creek bed in-between two specific points that the experimenters had marked (*Figure 3*). They traversed the trail three times in each direction, for a total of 6 traversals per subject. The Berkeley participants walked along a hiking trail in-between two distinctive landmarks. They traversed the trail once in each direction, for a total of 2 traversals per subject. The trail included pavement, flat terrain, medium terrain, and rough terrain, so as with the walk's start and end points, the experimenters used existing landmarks in the environment to mark the transitions between terrain types. We found the sections of recordings marked as rough terrain and included only that subset of the data in this study.

## Data processing

Following data collection, we performed a post hoc eye tracking calibration using the 9-point VOR calibration task data. Per subject, we placed a reference marker on the calibration fixation target at 10 timepoints in the recording (corresponding to the 9 points of the VOR calibration task, plus an additional repeat of the center marker at the end). With the Pupil Player app in natural features mode (Pupil Labs, Berlin, Germany), we used those markers to perform gaze mapping, generating 3D gaze vectors for both eyes.

We then had three sets of tracking data recorded at two different timescales and expressed in three different coordinate systems: the left eye's gaze data, the right eye's gaze data, and the 3D skeleton's pose data. Thus, our next step was to temporally and then spatially align the recordings via a procedure detailed in *Matthis et al., 2022*.

The timestamps from both systems were already synchronized to the same clock since they were recorded by the same computer, but the sampling rates (motion capture, 100 Hz; eye tracking 120 Hz) and specific timestamps were different. Using MATLAB's 'resample' function (Signal Processing Toolbox; MathWorks, Natick, MA, USA), we performed interpolation so that the motion capture and eye tracking data streams had the same sampling rate and timestamps (120 Hz). The result was that the left eye, right eye, and kinematic data streams were temporally aligned.

Once the three sets of data were temporally aligned, we used the VOR calibration data to spatially align them. During the VOR task, participants were fixating on a single point while moving their head, so we aligned each eye's coordinate system to that of the 3D skeleton by shifting and rotating them, such that the eyes were in an appropriate location relative to the head and the gaze vectors remained directed at the calibration fixation target as the head and eyes moved. To determine the shift per eye coordinate system, we estimated the position of each eye's center in 3D skeleton coordinates. We based our estimate on (a) the position and orientation of the skeleton's head node and (b) average measurements of the where the eyes are located within the human head. To determine the rotation per eye coordinate system, we found the single optimal rotation that minimized the distance between the calibration fixation target and the gaze vector's intersection with the mat. Note that because the head's position and orientation changed throughout the recording, we applied transformations relative to the position and orientation of the skeleton's head node in each frame.

Once the eye and body tracking data were fully aligned, we used the data from the body pose foot nodes to find the time and location of each step (both heel strike and toe off), following the velocity-based step-finding procedure outlined in *Zeni et al., 2008*.

Additionally, we identified periods of time when subjects were potentially collecting visual information by differentiating between when they were fixating and when they were making saccadic eye movements. We only used periods of fixation when considering the visual information available to participants, as mid-saccade visual input is unlikely to be used for locomotor guidance due to saccadic suppression and image blur.

We identified fixations by applying an eye-in-orbit velocity threshold of 65 °/s and an acceleration threshold of 5 °/s$^2$. (Note that the velocity threshold is quite high to avoid including the smooth counter-rotations that occur during eye stabilization). If both values were below threshold for a given frame, we classified that frame as containing a fixation; if not, the frame was classified as containing a saccade.

## Terrain reconstruction

To factor terrain height into our analyses, we needed information about the 3D structure of the terrain that participants walked over. Our dataset did not include data on terrain depth, but the scene videos recorded by the eye tracker's outward-facing camera did provide us with two-dimensional (2D) images of the terrain. In principle, photogrammetry should allow us to extract accurate 3D information about the terrain's structure from those 2D video frames, so our first step beyond the original analyses of these data (*Bonnen et al., 2021*; *Matthis et al., 2022*) was to use photogrammetry to generate 3D terrain reconstructions.

### Photogrammetry pipeline

To reconstruct the 3D environment from our 2D videos, we used an open-source software package called Meshroom, which is based on the AliceVision Photogrammetric Computer Vision framework (*Griwodz et al., 2021*). Meshroom combines multiple image processing and computer vision algorithms, ultimately allowing the user to estimate both environmental structure and relative camera position from a series of images.

The steps in the AliceVision photogrammetry pipeline are (1) natural feature extraction, (2) image matching, (3) features matching, (4) structure from motion, (5) depth maps estimation, (6) meshing, and (7) texturing (*Griwodz et al., 2021*).

To summarize in greater detail: (1) Features that are minimally variant with respect to viewpoint are extracted from each image. (2) To find images that show the same areas of the scene, images are grouped and matched on the basis of those features. (3) The features themselves are then matched between the two images in each candidate pair. (4) Those feature matches are then used to infer rigid scene structure (3D points) and image pose (position and orientation) via an incremental pipeline that operates on each image pair, uses the best pair to compute an initial two-view reconstruction, and then iteratively extends that reconstruction by adding new views. (5) The inferred 3D points are used to compute a depth value for each pixel in the original images. (6) The depth maps are then merged into a global octree, which is refined through a series of operations that ultimately produce a dense geometric surface representation of the scene. (7) Texture is added to each triangle in the resulting mesh via an approach that factors in each vertex's visibility and blends candidate pixel values with a bias toward low-frequency texture.

The most relevant outputs of this pipeline for our analyses are the 3D triangle mesh and the 6D camera trajectory (i.e. the estimated position and orientation of the camera, per input frame). We also used the textured triangle mesh but solely for visualization (e.g. *Figure 3*).

### Reconstruction procedure

Prior to terrain reconstruction, we processed the raw scene videos recorded by the eye tracker's outward-facing camera. We first used the software package 'ffmpeg' to extract the individual frames from the videos. We then undistorted each frame using a camera intrinsic matrix, which we estimated via checkerboard calibration (*Qilong and Pless, 2004*).

Then, we used Meshroom to process the scene video frames, one traversal at a time, specifying the camera intrinsics (focal length in pixels and viewing angle in degrees) and using Meshroom's

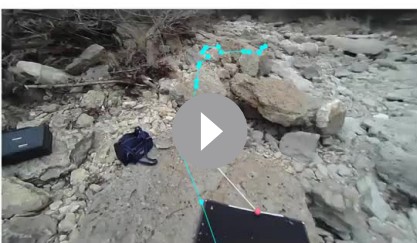

**Video 2.** Visualization of foothold locations in the scene camera's view for one traversal of the Austin trail: https://youtu.be/llulrzhlAVg. Computed foothold locations are marked with cyan dots.
https://elifesciences.org/articles/91243/figures#video2

default parameters. Meshroom processed the scene video frames according to the pipeline described above, producing both a terrain mesh and a 6D camera trajectory (3D position and 3D orientation), with one 6D vector for each frame of the original video. To give a sense of the mesh output, we have provided a rendered image of a small section of the textured Meshroom output in *Figure 1*.

## Data alignment

*Figure 2* illustrates the data alignment process which positions the body and eye movement data within the reconstructed terrain. Data alignment was performed on a per-traversal basis. First the body/eye tracking data was translated, pinning the location of the head node to Meshroom's estimated camera position. Next, the orientation of the head node was matched to Meshroom's estimated camera orientation by finding a single three-element Euler angle rotation that minimized the L2 error (i.e. the sum of squared errors) across frames using MATLAB's 'fminsearch' function. After applying that rotation, the body/eye tracking data was scaled so that, across all heel strikes in a given recording, the distance between the skeleton's heel and the closest point on the mesh at the time of that heel strike was minimized.

A visualization of the aligned motion capture, eye tracking, and terrain data for one traversal of the Austin trail can be seen in *Video 1*. We can also project locations into the scene camera video. *Video 2* shows an example of this, visualizing foothold locations in the scene camera's view for one traversal of the Austin trial.

## Evaluating terrain reconstruction

To evaluate the accuracy of the 3D reconstruction, we used the terrain meshes estimated from different traversals of the same terrain, both by an individual subject and also by the different subjects. We used only the Austin data here, as that dataset included 6 traversals per subject (vs. 2) and was collected over a much shorter time span (5 days) and thus was less likely to physically change.

The meshes were aligned using the open-source software package CloudCompare. To align two meshes in CloudCompare, one mesh needs to be designated as the fixed 'reference' mesh and the other as the moving 'aligned' mesh (i.e. the mesh that will be moved to align with the reference). We first coarsely aligned the meshes via a similarity transform. That step requires an initial set of keypoints,

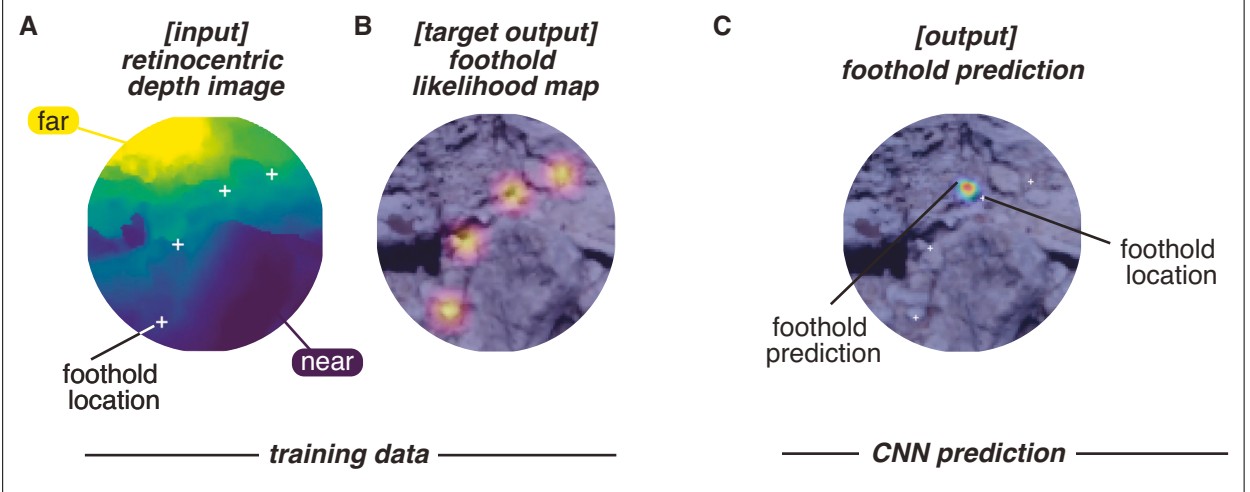

**Figure 11.** Convolutional neural network (CNN) inputs and outputs. Schematic shows the inputs and outputs for one example frame. (**A**) Input: retinocentric depth image. (**B**) Target output: foothold likelihood map. (**C**) Output: predicted foothold locations.

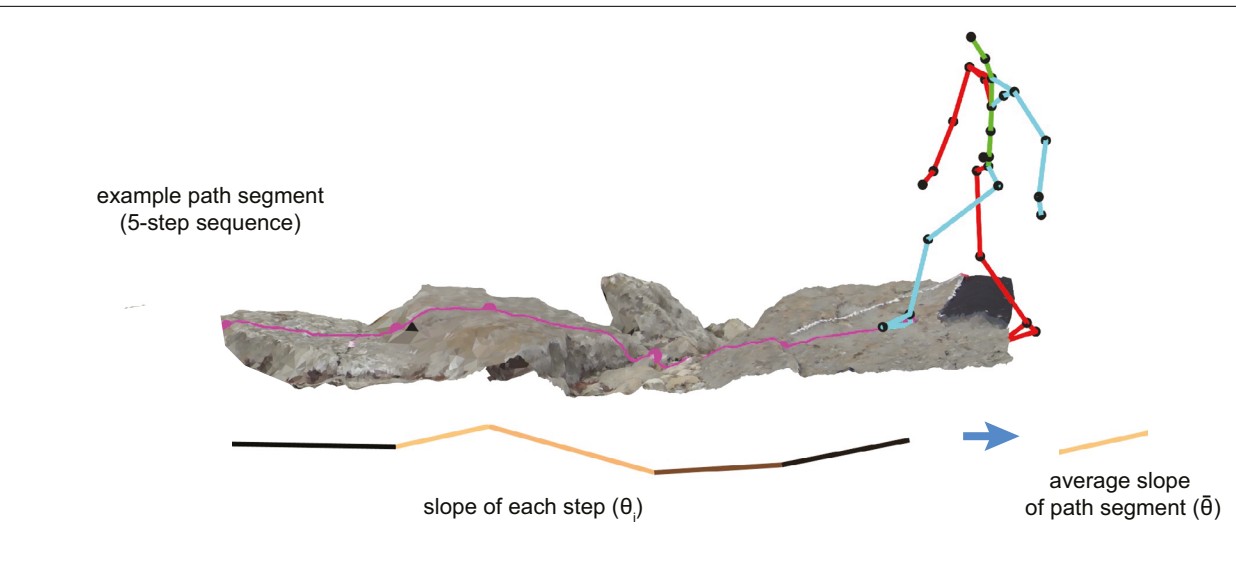

**Figure 12.** Schematic depicting the calculation of the mean slope of a step sequence. To calculate the slope of a step sequence – chosen or simulated – we first calculated the step slope for each step in the path, and we then averaged the absolute values of those slopes.

so we chose five easily discernible features in the environment (e.g. permanent marks on rocks) that were visible in each terrain mesh and manually marked their locations. We then completed the point cloud registration on a finer scale using the iterative closest point method. That procedure involves locating, for each point in the moving mesh, the closest point in the fixed point cloud (i.e. the moving point's nearest neighbor). The distance between nearest neighbors is then iteratively minimized via best-fitting similarity transforms.

We can then evaluate the reliability of terrain reconstruction by measuring the distance between nearest neighbors on the two meshes. We make two key comparisons: (1) measuring nearest neighbor distances for all points on the terrain mesh within 2 m of the path (see *Figure 5A*) and (2) measuring nearest neighbor distances for all footholds locations by taking the nearest neighbor distance for the mesh point closest to that foothold (see *Figure 5B*). For the path comparison, the median error was 4.5 cm, and the 95% quantile was 20.0 cm (*Figure 5A*). For the foothold comparison, the median error was 4.0 cm, and the 95% quantile was 16.8 cm (*Figure 5B*). To put this into context, the average foot of a person from North America is 25.8 cm (*Jurca et al., 2019*). In both cases, the majority of mesh errors fall below 20% of the average foot size.

## Retinocentric analysis

To assess whether depth features can be used to explain some variation in foothold selection, we trained a CNN to predict future foothold locations given the walker's view of the terrain's depth. This analysis involved computing both the retinocentric depth images that served as the input data (*Figure 11A*) and the foothold likelihood maps that served as target output for training (*Figure 11B*). The retinocentric depth images approximate the visual information subjects have when deciding on future foothold locations, and the foothold likelihood maps represent the subjects' subsequent decisions. After training and testing the CNN, we quantified its prediction accuracy by calculating the AUC. With that metric, scores above chance (50%) would indicate that the depth information plays some role in determining where individuals will put their feet.

### Retinocentric depth images

We computed subject-perspective depth images (e.g. *Figure 11A*) using the aligned eye tracking, motion capture, and photogrammetry data in the open-source 3D computer graphics software Blender. Per subject, we moved a virtual camera through the reconstructed terrain, updating its position and orientation in each frame based on Meshroom's estimate of the scene camera's location in the corresponding frame of the scene video. Blender's 'Z-buffer' method then captured an image showing the depth of the 3D triangle mesh representation relative to the camera.

**Table 2.** Layers of custom convolutional neural network (CNN).

| Layer | Output shape | # Params |
|---|---|---|
| Conv2D | (100, 100, 4) | 404 |
| BatchNormalization | (100, 100, 4) | 16 |
| MaxPooling2D | (50, 50, 4) | 0 |
| Conv2D | (50, 50, 8) | 3208 |
| BatchNormalization | (50, 50, 8) | 32 |
| MaxPooling2D | (25, 25, 8) | 0 |
| Conv2D | (25, 25, 16) | 12,816 |
| BatchNormalization | (25, 25, 16) | 64 |
| Conv2DTranspose | (25, 25, 16) | 25,616 |
| BatchNormalization | (25, 25, 16) | 64 |
| UpSampling2D | (50, 50, 16) | 0 |
| Conv2DTranspose | (50, 50, 8) | 12,808 |
| BatchNormalization | (50, 50, 8) | 32 |
| UpSampling2D | (100, 100, 8) | 0 |
| Conv2DTranspose | (100, 100, 4) | 3204 |
| Conv2DTranspose | (100, 100, 1) | 401 |
| BatchNormalization | (100, 100, 1) | 4 |
| Flatten | (10,000) | 0 |
| Softmax | (10,000) | 0 |
| Reshape | (100, 100) | 0 |

The resulting egocentric depth images were then transformed to polar coordinates (polar angle $\theta$ and eccentricity $\rho$) to approximate a retinocentric perspective. We defined the diameter of images as 45° of visual angle, and we used 2D interpolation to compute the pixel values at each polar coordinate from the pixel values in Cartesian coordinates.

The depth values in the images were then converted into relative depth values. To make that shift, we subtracted the gaze point's depth (i.e. the value at the center pixel) from the entire depth image. The value at center thus became 0, and the rest of the depth values were relative to the depth of the gaze point.

## Foothold likelihood maps

To calculate the ground truth of the future foothold locations in each depth image, we found the point at which the line between the current camera position and the foothold intersected the camera's image plane.

The ground truth foothold locations in the world video frame were converted to likelihood maps (e.g. *Figure 11B*) by smoothing foothold locations with a Gaussian kernel: $\sigma$ = 5 pixels. (In degrees of visual angle, the kernel size was roughly 1°. That value is not exact because the conversion between pixels and degrees is not constant throughout the visual field). This smoothing mitigated the impact of any noise in our estimation of foothold location to allow more robustness in the CNN learned features.

## Convolutional neural network

The retinocentric depth images and foothold likelihood maps were then used to train a custom CNN to predict the probability that each location in the retinocentric depth images was a foothold location. The network input was a depth image (*Figure 11A*), and the target output was a foothold likelihood map (*Figure 11B*).

The CNN had a convolutional-deconvolutional architecture with three convolutional layers followed by three transposed convolutional layers (*Table 2*). Training was performed using KL divergence between the CNN output (*Figure 11C*) and the foothold likelihood maps (*Figure 11B*). Data was split so that half of the pairs of depth images and likelihood maps were used to train the network and the other half was reserved for testing. This split ensured that the network was tested on terrain that it had not previously 'seen'.

To evaluate performance, we calculated the AUC per depth image. The true foothold locations per image were known, and the CNN generated a probability per pixel per image. To generate the ROC curve, we treated the CNN task as a binary classification of pixels, and we calculated the rate of false positives and true positives at different probability criterion values, increasing from 0 to 1. Calculating AUC was then just a matter of computing the area under the resulting ROC curve.

## Step analysis

We sought to better understand how subjects chose their footholds by analyzing the properties of their chosen steps and step sequences. Throughout this work, a foothold location is defined as the 3D position of the left or right foot marker at the time of heel strike, and a step is defined as the transition between two footholds. To analyze sets of steps, we segmented participants' paths into 5-step sequences, consisting of 6 consecutive footholds.

### Step properties

For each step in the dataset, we computed seven properties: distance, ground distance, direction, goal direction, deviation from goal, height, and slope.

To illustrate how we compute a step's properties, consider a step vector $\vec{s}$ that starts at a foothold with coordinates $(x_1, y_1, z_1)$ and ends at a foothold with coordinates $(x_2, y_2, z_2)$, where the $y$-axis corresponds to gravity.

### In 3D

We define step distance $D$ as the magnitude of step vector $\vec{s}$, i.e., the 3D Euclidean distance between the start and end footholds:

$$D = |\vec{s}| = \sqrt{(x_2 - x_1)^2 + (y_2 - y_1)^2 + (z_2 - z_1)^2} \tag{1}$$

### In 2D, from overhead

To focus on the progression of the step along the subject's route and ignore the step's vertical component, we project step vector $\vec{s}$ onto the ground plane ($xz$-space), producing ground vector $\vec{g}$. We define step ground distance $G$ as the magnitude of ground vector $\vec{g}$, i.e., the 2D Euclidean distance in $xz$-space between the start and end footholds:

$$G = |\vec{g}| = \sqrt{(x_B - x_A)^2 + (z_B - z_A)^2} \tag{2}$$

We define step direction $\gamma$ as the direction of ground vector $\vec{g}$:

$$\gamma = \arctan \frac{z_B - z_A}{x_B - x_A} \tag{3}$$

We then consider the end point of the current terrain traversal, foothold $E$. Along the ground plane, the vector $\vec{e}$ connects the step's starting foothold $(x_A, z_A)$ to that traversal's end point $(x_E, z_E)$. That vector represents the most direct path the participant could take to reach their current goal. We refer to the direction of vector $\vec{e}$ as the goal direction $\omega$:

$$\omega = \arctan \frac{z_E - z_A}{x_E - x_A} \tag{4}$$

We use that angle to calculate step deviation from goal $\delta$, which we define as the angle between the step direction $\gamma$ and goal direction $\omega$:

$$\delta = \gamma - \omega \tag{5}$$

### In 2D, from the side

To analyze the step's vertical component, we calculate step height $\Delta h$ by finding the change in vertical position between footholds $A$ and $B$:

$$\Delta h = y_B - y_A \tag{6}$$

We then compute step slope by dividing step height $\Delta h$ by step distance $D$:

$$\theta = \arcsin \frac{\Delta h}{D} \tag{7}$$

### Properties of step sequences

Step sequences (sometimes also called paths or path segments) are composed of a series of steps. In this paper, we primarily focused on 5-step sequences. We calculated two key properties: mean slope and tortuosity.

### Mean slope

Each step sequence has a mean slope, $\bar{\theta}$, which is defined as the average step slope across steps within the sequence (*Figure 12*):

$$\bar{\theta} = \frac{\sum_i^n \theta_i}{n} \tag{8}$$

### Tortuosity

We quantified the curvature of each step sequence by calculating tortuosity, $T$:

$$T = \frac{\sum_i^n D_i}{D_s} \tag{9}$$

where $n$ is the number of steps in the sequence, $D_i$ is the magnitude of a given step vector, and $D_s$ is the magnitude of the vector connecting the start and end foothold locations.

Thus, we quantify tortuosity as the ratio of the cumulative step distance to the distance between the first and final footholds. This metric is the inverse of the straightness index formula proposed in *Batschelet, 1981*, which has been shown to be a reliable estimate of the tortuosity of oriented paths (*Benhamou, 2004*). A tortuosity of 1 indicates a straight path, while a tortuosity greater than 1 indicates a curved path. A perfect semi-circle would have a tortuosity of $\pi/2$ (approximately 1.57), and a circle would be infinitely tortuous.

## Path simulation

We sought to evaluate differences between the paths subjects chose and the alternative paths they could have chosen. To do so, we simulated 5-step sequences and compared the properties of chosen paths to those of simulated paths.

### Identifying viable footholds

Previous work has found that subjects are able to walk on surfaces slanted up to approximately 33° (*Kinsella-Shaw et al., 1992*). We thus constrained possible foothold locations to those with a local surface slant below that value.

To calculate the slant of possible footholds, we computed the surface normal vector for each triangle in the 3D terrain mesh. We then calculated the mean local surface slant for each point in the mesh's point cloud representation by averaging the surface slants of all triangles within a radius of 1 foot length (25.8 cm).

### Identifying viable steps

After identifying viable foothold locations, viable steps between viable foothold locations were determined based on three constraints (*Figure 7*). Per subject, we found the distributions of (a) step slope (*Equation 7*), (b) step ground distance (*Equation 2*) relative to that participant's leg length, and (c)

step deviation from goal (*Equation 5*). We computed the maximum observed absolute values, and we then deemed a step viable only if its properties fell within those maxima.

## Simulating possible paths

### Paths with a fixed start point and a random end point

For each foothold that a subject chose, we simulated possible alternative paths consisting of 5 steps (i.e. 6 footholds). These simulated path segments started at the chosen foothold, and subsequent footholds were iteratively selected, in accordance with the foothold constraint and step constraints defined above. The resulting set of path segments could then be directly compared to the path segment that the walker actually chose.

### Paths with fixed start and end points

We also simulated paths that started and ended at chosen foothold locations. To accomplish that, we treated the set of possible footholds and viable steps between them as a directed graph. We then used MATLAB's 'maxflow' function to find the subset of footholds that have non-zero flow values in a directed graph between the two selected footholds (starting point and ending point). The 'maxflow' function then returns a set of footholds that can be visited from the starting foothold and still have available paths to the final foothold (i.e. 6th foothold in path).

Possible paths connecting the two end points of the actual path are then generated from this subset of possible foothold locations following the procedure in the previous section, iteratively selecting footholds in accordance with the step constraints defined above.

## Estimating straight path slope

When walking from one point to another in flat terrain, straight paths are almost certainly the preferable option. In rough terrain, however, there may be obstacles that make walking straight impossible – or at least less preferable – than taking a slightly longer curved path. To analyze this potential trade-off, for each step sequence, we estimated the slope a walker would encounter if they tried to take a relatively straight path.

To compute those values, we first found the median tortuosity of all chosen 5-step sequences in a particular terrain traversal (i.e. across a particular mesh). That gave us a conservative, terrain-specific tortuosity threshold that we could use to determine which of the possible paths were relatively straight. For each step sequence in that traversal, we then identified simulated paths with a tortuosity below that threshold and calculated the average of their mean step slopes ($\bar{\theta}$; *Equation 8*). The resulting values are treated as the average step slope the subject would encounter if they tried to take a straighter path for that segment of terrain.

## Acknowledgements

This work was supported by NIH Grants EY05729 and K99 EY 028229.

## Additional information

### Funding

| Funder | Grant reference number | Author |
| --- | --- | --- |
| National Institutes of Health | R01 EY05729 | Mary M Hayhoe |
| National Institutes of Health | T32 LM012414 | Karl S Muller |
| National Institutes of Health | K99 EY028229 | Jonathan Matthis |

The funders had no role in study design, data collection and interpretation, or the decision to submit the work for publication.

## Author contributions
Karl S Muller, Conceptualization, Data curation, Software, Formal analysis, Validation, Investigation, Visualization, Methodology, Writing – original draft, Writing – review and editing; Kathryn Bonnen, Data curation, Formal analysis, Validation, Investigation, Visualization, Writing – review and editing; Stephanie M Shields, Validation, Writing – review and editing; Daniel P Panfili, Conceptualization, Methodology; Jonathan Matthis, Conceptualization, Software, Methodology; Mary M Hayhoe, Conceptualization, Resources, Supervision, Funding acquisition, Investigation, Methodology, Writing – original draft, Project administration, Writing – review and editing

## Author ORCIDs
Kathryn Bonnen https://orcid.org/0000-0002-9210-8275
Jonathan Matthis https://orcid.org/0000-0003-3683-646X
Mary M Hayhoe https://orcid.org/0000-0002-6671-5207

## Ethics
Human subjects: Human subjects: Informed consent and consent to publish was obtained and protocols were approved by the institutional IRBs at University of Texas Austin approval number 2006- 06- 0085 and UC Berkeley approval number 2011- 07- 3429.

Reviewer #1 (Public review): https://doi.org/10.7554/eLife.91243.3.sa1
Reviewer #2 (Public review): https://doi.org/10.7554/eLife.91243.3.sa2
Reviewer #3 (Public review): https://doi.org/10.7554/eLife.91243.3.sa3
Author response https://doi.org/10.7554/eLife.91243.3.sa4

# Additional files

## Supplementary files
• MDAR checklist

## Data availability
Data and code for generating *Figures 5, 7–10*, *Figure 7—figure supplements 1–3*, and *Figure 9—figure supplements 1 and 2* has been made available via Dryad and Zenodo.

The following datasets were generated:

| Author(s) | Year | Dataset title | Dataset URL | Database and Identifier |
|---|---|---|---|---|
| Muller K, Panfili D, Shields S, Matthis J, Bonnen K, Hayhoe M | 2024 | Foothold selection during locomotion in uneven terrain: Results from the integration of eye tracking, motion capture, and photogrammetry | https://doi.org/10.5061/dryad.r7sqv9sn2 | Dryad Digital Repository, 10.5061/dryad.r7sqv9sn2 |
| Muller K, Panfili D, Shields S, Matthis J, Bonnen K, Hayhoe M | 2024 | Foothold selection during locomotion in uneven terrain: Results from the integration of eye tracking, motion capture, and photogrammetry | https://doi.org/10.5281/zenodo.13932346 | Zenodo, 10.5281/zenodo.13932346 |

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
