## [Editor Report · eLife assessment]

This **fundamental** study has the potential to substantially advance our understanding of human locomotion in complex real-world settings and opens up new approaches to studying (visually guided) behavior in natural settings outside the lab. The evidence supporting the conclusions is overall **compelling**. Whereas detailed analyses represent multiple ways to visualize and quantify the rich and complex natural behavior, some of the specific conclusions remain more suggestive at this point. The work will be of interest to neuroscientists, kinesiologists, computer scientists, and engineers working on human locomotion.

---

## [Referee Report · Reviewer #1 (Public review)]

Summary:

The work of Muller and colleagues concerns the question where we place our feet when passing uneven terrain, in particular how we trade-off path length against the steepness of each single step. The authors find that paths are chosen that are consistently less steep and deviate from the straight line more than an average random path, suggesting that participants indeed trade off steepness for path length. They show that this might be related to biomechanical properties, specifically the leg length of the walkers. In addition, they show using a neural network model that participants could choose the footholds based on their sensory (visual) information about depth.

Strengths:

The work is a natural continuation of some of the researchers' earlier work that related the immediately following steps to gaze. Methodologically, the work is very impressive and presents a further step forward towards understanding real-world locomotion and its interaction with sampling visual information. While some of the results may seem somewhat trivial in hindsight (as always in this kind of studies), I still think this is a very important approach to understand locomotion in the wild better.

Weaknesses:

The concerns I had regarding the initial version of the manuscript have all been fixed in the current one.

---

## [Referee Report · Reviewer #2 (Public review)]

This manuscript examines how humans walk over uneven terrain and use vision to decide where to step. There is a huge lack of evidence about this because the vast majority of locomotion studies have focused on steady, well-controlled conditions, and not on decisions made in the real world. The author team has already made great advances in this topic by pioneering gaze recordings during locomotion, but there has been no practical way to map the gaze targets, specifically the 3D terrain features in naturalistic environments. The team has now developed a way to integrate such measurements along with gaze and step tracking. This allows quantitative evaluation of the proposed trade-offs between stepping vertically onto vs. stepping around obstacles, along with how far people look to decide where to step. The team also introduces several new analysis techniques to accompany these measurements. They use machine learning techniques to examine whether retinocentric depth helps predict footholds and develop simulations to assess possible alternative footholds and walking paths. The technical achievement is impressive.

This study addresses several real-world questions not normally examined in the laboratory. First, do humans elect to walk around steeper footholds rather than over them? Second, is there a quantifiable benefit to walking around, such as allowing for a flatter path? Third, does visual depth of terrain contribute to selection of footholds? Fourth, are there scale effects, where for example a tall adult can easily walk over an obstacle that a toddler must walk around. One might superficially answer yes to all of these questions, but it is highly nontrival to answer them quantitatively. As for the conclusions, my feelings are mixed. I find strengths in answers to two of the questions, and weaknesses in the other two.

Strengths:

I consider the evidence strongest for the first of the main questions. The results show subjects walking with more laterally deviating paths, measured by a quantity called "tortuosity," when the direct straight-ahead paths appear to have steeper ups and downs (Fig. 9). The measure of straight-ahead steepness is fairly complicated (discussed below), but is shown to be well correlated with tortuosity, effectively predicting when subjects will not walk straight ahead.

There is also good evidence for the third question, showing that retinocentric depth is predictive of chosen footholds. Retinocentric depth was computed by a series of steps, starting with scene capture to determine a 3D terrain mesh, projecting that mesh into the eye's perspective, and then discarding all but the depth information. This highly involved process is only the beginning, because the depth was then used to train a neural network classifier with chosen footholds. That network was found to predict footholds better than chance, using a test set independent from the training set, each using half the recorded data. The results are strong and are best interpreted along with a previous study (Bonnen et al. 2021) showing that subjects gaze nearer ahead on rougher terrain, and slightly more so when binocular vision was disrupted. Depth information seems important for foothold selection.

As an aside, humans presumably also select footholds and estimate depth from a number of monocular visual cues, such as shading, shadows, color, and self-motion information. Interestingly, the terrain mesh and depth data here were computed from monocular images, suggesting that monocular vision can in principle be predictive of both depth and footholds. Binocular human vision presumably improves on monocular depth estimation, and so it would be interesting to see whether binocular scene cameras would predict footholds better. In an earlier review, I had suggested other avenues for exploration, but these are not weaknesses so much as opportunities not yet taken. I believe much could be learned from deeper analysis of the neural network, and future experiments using variations of this technique.

There is much to be appreciated about this study. I was impressed by the overarching outlook and ambitiousness of the team. They seek to understand human decision-making in real-world locomotion tasks, a topic of obvious relevance to the human condition but not often examined in research. The field has been biased toward well-controlled, laboratory studies, which have undeniable scientific advantages but are also quite different from the real world. The present study discards all of the usual advantages of the laboratory, yet still finds a way to explore real-world behaviors in a quantitative manner. It is an exciting and forward-thinking approach, used to tackle an ecologically relevant question.

I also appreciate the numerous technical challenges of this study. The state of the art in real-world locomotion studies has largely been limited to kinematic motion capture. This team managed to collect and analyze an unprecedented, one-of-a-kind dataset. They applied a number of non-trivial methods to assess retinocentric depth, simulate would-be walking paths and steepness, and predict footholds from neural network. Any of these could and probably will merit individual papers, and to assemble them all at once is quite beyond other studies I am aware of. I hope this study will spur more inquiries of this type, leveraging mobile electronics and modern machine learning techniques to answer questions that were previously only addressable qualitatively.

Weaknesses:

Although I am highly enthusiastic about this study, I was not entirely convinced by the evidence for the second and fourth questions. Some of this is because I was confused by aspects of the analysis, limiting my understanding of the evidence. But I also question some of the basic conclusions, whether the authors indeed proved that (from Abstract, emphasis mine) "[walkers] change direction TO AVOID taking steeper steps that involve large height changes, instead of [sic] choosing more circuitous, RELATIVELY FLAT paths." (I interpret the "of" as a typo that should have been omitted). I think it is more objective to say, "walkers changed direction more when straight-ahead paths seemed to have steeper height changes."

I say "seemed" because it is unknown whether humans would have experienced greater height changes if they walked straight ahead (the second main question). The comparison shown is between human tortuous paths taken and simulated straight-ahead paths never experienced by human. Ignoring questions about the simulations for now (discussed below), it is not an apples-to-apples comparison, say between the tortuous paths humans preferred and straight-ahead paths they didn't. The authors determined a measure of steepness, "straight path slope" (Fig. 9), that predicts when humans circuitously, but that is the same as the steepness that humans would actually experience if they had walked straight ahead. That could have been measured with an appropriate control condition, for example asking subjects to walk as straight ahead as they can manage. That also would have eliminated the need for simulations, because the slope of each step actually taken could simply have been measured and compared between conditions. Instead, two different kinds of simulations are compared, where steeper paths are fully simulated, and the circuitous paths are partially simulated but partially based on data. It seems that every fifth circuitous step coincides with a human foothold, but the intervening ones are somewhat random. I don't find this especially strong evidence that the chosen paths were indeed relatively flatter. I would prefer to be convinced by hard data than by unequal simulations.

I also have trouble accepting "TO AVOID" because it implies a degree of intent not evident in the data. I suppose conscious intent could be assessed subjectively by questionnaire, but I don't know how unconscious intent could be tested objectively. I believe my suggested interpretation above is better supported by evidence.

My limited acceptance is due in part to confusion about the simulations. I was especially confused about the connection between feasible steps drawn from the distribution in Figure 7, and the histograms of Figure 8. The feasible steps have clear peaks near zero slope, unity step length, and zero step direction (let's call them Flat). If 5-step simulations of Figure 8 draw from that distribution, why is there zero probability for the 0-3 deg bin (which is within {plus minus}3 deg due to absolute values)? It seems to me that Flat steps were eminently available, so why were they completely avoided? It seems that the simulations were probabilistic (and not just figurative) random walks, which implies they should have had about the same mean as Figure 7 but a wider variance, and then passed through absolute value. They look like something else that I cannot understand. This is important because the RELATIVELY FLAT conclusion is based on the chosen walks apparently being skewed flatter than random simulated walks. I have trouble accepting those distributions because Flat steps were unaccountably never taken by either simulation or human. (This issue is less concerning for Figure 9, because one can accept that some simulation measure is predictive of tortuosity even if the measure is hard to understand).

I was also confused why Figure 7 distances and directions are nearly normally distributed and not more uniform. The methods only mention constraints to eliminate steps, which to me suggests a truncated uniform distribution. It is not clear to me why the terrain should have a high peak at unity step length, which implies that the only feasible footholds were almost exclusively straight ahead and one step length away. It is possible that the "feasible" footholds are themselves drawn from a "likely" normal distribution, perhaps based on level walking data. It could be argued that simulated steps should be performed by drawing from typical step distributions for level ground, eliminating non-viable footholds, and then repeating that across multiple steps. That would explain the normality, but it is not stated in the Methods, and even if they were "feasible and likely" it would not explain the distributions of Figure 8.

I had some misgivings about the fourth question, where Figure 10 suggests that shorter subjects had greater correlation between straight-path slope and tortuosity than taller ones, who tended to walk straighter ahead. I agree with the authors' rebuttal to my previous review that "the data are the data" but I still have doubts. Now supplied as suggested by another reviewer, Figure 18 provides more detail of the underlying data, with considerably lower correlations. I now suspect that Figure 10 benefits from some statistical artifacts due to binning and other operations, and the weaker correlations of Fig. 18A are closer to reality. I am rather suspicious of correlations of correlations (Figure 18B), which lose some statistical grounding because the second correlation treats all data on equal footing, effectively whitewashing the first correlations of their varying significance (p-values 0.008 to 1e-9).

Furthermore, I am also unsure about Figure 10's comparison of tortuosity vs. straight path slope against leg length. Both tortuosity and straight path slope are already effectively dimensionless and therefore already seem to eliminate scale. It is my understanding that the simulated paths were recomputed for each subject's parameters, and the horizontal axis, slope, is already an angular measure that should affect short and tall people similarly. Shouldn't all subjects equally avoid steep angles, regardless of their dimensional height? If there is indeed a scale effect, then I would expect it to be demonstrated with a dimensional measure (vertical axis) that depends on leg length.

I certainly agree with the hypothetical prior that tall adults walk straight over obstacles that shorter adults (or children) walk around. But I feel that simpler tests would better evidence, perhaps in future work. Did shorter subjects walk with greater tortuosity than taller ones on the same terrain? Did shorter subjects take relatively more steps even after normalizing for leg length? A possible comparison would be (number of steps)*(leg length)/(start to end distance). I feel that the evidence from this study is not that strong.

Although it is a strength of this study that so much can be learned from pure observation, that does not mean controlled conditions are not scientifically helpful. As mentioned earlier, a helpful control could have been to ask subjects to walk straighter but less preferred paths on the same terrain, treating human paths as an independent variable. Another would be to treat terrain as an independent variable, by using level ground and intermediate terrain conditions. This would make it easier to test whether taller subjects walk straighter ahead on more uneven terrain than shorter subjects. Indeed, the data set already includes some patches of flatter terrain, not included here. Additional and simpler tests might be possible based on existing data.

Conclusion

This is an ambitious undertaking, presenting a wealth of unprecedented data to quantitatively test basic ecological questions that have long been unanswered. There are a number of considerable strengths that merit appreciation, especially the ability to quantitatively predict when humans will walk more circuitously. The weaknesses are about limitations in the conclusions that can be drawn thus far rather than the correctness of the study. I consider this to be a first step that will hopefully enable and inspire a long line of future work that will address these questions more in depth.

---

## [Referee Report · Reviewer #3 (Public review)]

Summary:

The systematic way in which path selection is parametrically investigated is the main contribution.

Strengths:

The authors have developed an impressive workflow to study gait and gaze in natural terrain. They are able to determine footholds and gaze points in the 3D world, and explore different path selections in the terrain.

Weaknesses:

The finding that walkers prefer less tortuous, demanding paths is hardly surprising, and from the data it is still not clear what actual visual features are used to choose among alternative routes or what the nature of the decision process is. The authors discuss energetic cost and other "factors" that might influence path selection, but as yet there is no way to express these ideas rigorously in such complex natural settings.

---

## [Author Response]

The following is the authors’ response to the original reviews.

We thank the reviewers for their constructive reviews. Taken together, the comments and suggestions from reviewers made it clear that we needed to focus on improving the clarity of the methods and results. We have revised the manuscript with that in mind. In particular, we have restructured the results to make the logic of the manuscript clearer and we have added details to the methods section.

**Public Reviews:**

**Reviewer #1 (Public Review):**
Summary:The work of Muller and colleagues concerns the question of where we place our feet when passing uneven terrain, in particular how we trade-off path length against the steepness of each single step. The authors find that paths are chosen that are consistently less steep and deviate from the straight line more than an average random path, suggesting that participants indeed trade-off steepness for path length. They show that this might be related to biomechanical properties, specifically the leg length of the walkers. In addition, they show using a neural network model that participants could choose the footholds based on their sensory (visual) information about depth.Strengths:The work is a natural continuation of some of the researchers' earlier work that related the immediately following steps to gaze [17]. Methodologically, the work is very impressive and presents a further step forward towards understanding real-world locomotion and its interaction with sampling visual information. While some of the results may seem somewhat trivial in hindsight (as always in this kind of study), I still think this is a very important approach to understanding locomotion in the wild better.Weaknesses:The manuscript as it stands has several issues with the reporting of the results and the statistics. In particular, it is hard to assess the inter-individual variability, as some of the data are aggregated across individuals, while in other cases only central tendencies (means or medians) are reported without providing measures of variability; this is critical, in particular as N=9 is a rather small sample size. It would also be helpful to see the actual data for some of the information merely described in the text (e.g., the dependence of \Delta H on path length). When reporting statistical analyses, test statistics and degrees of freedom should be given (or other variants that unambiguously describe the analysis).

There is only one figure (Figure 6) that shows data pooled over subjects and this is simply to illustrate how the random paths were calculated. The actual paths generated used individual subject data. We don’t draw our conclusions from these histograms – they are instead used to generate bounds for the simulated paths. We have made clear both in the text and in the figure legends when we have plotted an example subject. Other plots show the individual subject data. We have given the range of subject medians as well as the standard deviation for data illustrated in Figure (random vs chosen), we have also given the details of the statistical test comparing the flatness of the chosen paths versus the randomly generated paths. We have added two supplemental figures to show individual walker data more directly: (Fig. 14) the per subject histograms of step parameters, (Fig. 18) the individual subject distributions for straight path slopes and tortuosity.

The CNN analysis chosen to link the step data to visual sampling (gaze and depth features) should be motivated more clearly, and it should describe how training and test sets were generated and separated for this analysis.

We have motivated the CNN analysis and moved it earlier in the manuscript to help clarify the logic the manuscript. Details of the training and test are now provided, and the data have been replotted. The values are a little different from the original plot after making a correction in the code, but the conclusions drawn from this analysis are unchanged. This analysis simply shows that there is information in the depth images from the subject’s perspective that a network can use to learn likely footholds. This motivates the subsequent analysis of path flatness.

There are also some parts of figures, where it is unclear what is shown or where units are missing. The details are listed in the private review section, as I believe that all of these issues can be fixed in principle without additional experiments.

Several of the Figures have been replotted to fix these issues.

**Reviewer #2 (Public Review):**
Summary:This manuscript examines how humans walk over uneven terrain using vision to decide where to step. There is a huge lack of evidence about this because the vast majority of locomotion studies have focused on steady, well-controlled conditions, and not on decisions made in the real world. The author team has already made great advances in this topic, but there has been no practical way to map 3D terrain features in naturalistic environments. They have now developed a way to integrate such measurements along with gaze and step tracking, which allows quantitative evaluation of the proposed trade-offs between stepping vertically onto vs. stepping around obstacles, along with how far people look to decide where to step.Strengths:(1) I am impressed by the overarching outlook of the researchers. They seek to understand human decision-making in real-world locomotion tasks, a topic of obvious relevance to the human condition but not often examined in research. The field has been biased toward well-controlled studies, which have scientific advantages but also serious limitations. A well-controlled study may eliminate human decisions and favor steady or periodic motions in laboratory conditions that facilitate reliable and repeatable data collection. The present study discards all of these usually-favorable factors for rather uncontrolled conditions, yet still finds a way to explore real-world behaviors in a quantitative manner. It is an ambitious and forward-thinking approach, used to tackle an ecologically relevant question.(2) There are serious technical challenges to a study of this kind. It is true that there are existing solutions for motion tracking, eye tracking, and most recently, 3D terrain mapping. However most of the solutions do not have turn-key simplicity and require significant technical expertise. To integrate multiple such solutions together is even more challenging. The authors are to be commended on the technical integration here.(3) In the absence of prior studies on this issue, it was necessary to invent new analysis methods to go with the new experimental measures. This is non-trivial and places an added burden on the authors to communicate the new methods. It's harder to be at the forefront in the choice of topic, technical experimental techniques, and analysis methods all at once.Weaknesses:(1) I am predisposed to agree with all of the major conclusions, which seem reasonable and likely to be correct. Ignoring that bias, I was confused by much of the analysis. There is an argument that the chosen paths were not random, based on a comparison of probability distributions that I could not understand. There are plots described as "turn probability vs. X" where the axes are unlabeled and the data range above 1. I hope the authors can provide a clearer description to support the findings. This manuscript stands to be cited well as THE evidence for looking ahead to plan steps, but that is only meaningful if others can understand (and ultimately replicate) the evidence.

We have rewritten the manuscript with the goal of clarifying the analyses, and we have re-labelled the offending figure.

(2) I wish a bit more and simpler data could be provided. It is great that step parameter distributions are shown, but I am left wondering how this compares to level walking. The distributions also seem to use absolute values for slope and direction, for understandable reasons, but that also probably skews the actual distribution. Presumably, there should be (and is) a peak at zero slope and zero direction, but absolute values mean that non-zero steps may appear approximately doubled in frequency, compared to separate positive and negative. I would hope to see actual distributions, which moreover are likely not independent and probably have a covariance structure. The covariance might help with the argument that steps are not random, and might even be an easy way to suggest the trade-off between turning and stepping vertically. This is not to disregard the present use of absolute values but to suggest some basic summary of the data before taking that step.

We have replotted the step parameter distributions without absolute values. Unfortunately, the covariation of step parameters (step direction and step slope) is unlikely to help establish this tradeoff. Note that the primary conclusion of the manuscript is that works make turns to keep step slope low (when possible). Thus, any correlation that might exist between goal direction and step slope would be difficult to interpret without a direct comparison to possible alternative paths (as we have done in this paper). As such we do not draw our conclusions from them. We use them primarily to generate plausible random paths for comparison with the chosen paths. We have added two supplementary figures including distributions (Fig 15) and covariation of all the step parameters discussed in the methods (Fig 16).

(3) Along these same lines, the manuscript could do more to enable others to digest and go further with the approach, and to facilitate interpretability of results. I like the use of a neural network to demonstrate the predictiveness of stepping, but aside from above-chance probability, what else can inform us about what visual data drives that?

The CNN analysis simply shows that the information is there in the image from the subject’s viewpoint and is used to motivate the subsequent analysis. As noted above, we have generally tried to improve the clarity of the methods.

Similarly, the step distributions and height-turn trade-off curves are somewhat opaque and do not make it easy to envision further efforts by others, for example, people who want to model locomotion. For that, clearer (and perhaps) simpler measures would be helpful.

We have clarified the description of these plots in the main text and in the methods. We have also tried to clarify why we made the choices that we did in measuring the height-turn trade-off and why it is necessary in order to make a fair comparison.

I am absolutely in support of this manuscript and expect it to have a high impact. I do feel that it could benefit from clarification of the analysis and how it supports the conclusions.
**Reviewer #3 (Public Review):**
Summary:The systematic way in which path selection is parametrically investigated is the main contribution.Strengths:The authors have developed an impressive workflow to study gait and gaze in natural terrain.Weaknesses:(1) The training and validation data of the CNN are not explained fully making it unclear if the data tells us anything about the visual features used to guide steering. It is not clear how or on what data the network was trained (training vs. validation vs. un-peeked test data), and justification of the choices made. There is no discussion of possible overfitting. The network could be learning just e.g. specific rock arrangements. If the network is overfitting the "features" it uses could be very artefactual, pixel-level patterns and not the kinds of "features" the human reader immediately has in mind.

The CNN analysis has now been moved earlier in the manuscript to help clarify its significance and we have expanded the description of the methods. Briefly, it simply indicates that there is information in the depth structure of the terrain that can be learned by a network. This helps justify the subsequent analyses. Importantly, the network training and testing sets were separated by terrain to ensure that the model was being tested on “unseen” terrain and avoid the model learning specific arrangements. This is now clarified in the text.

(2) The use of descriptive terminology should be made systematic.Specifically, the following terms are used without giving a single, clear definition for them: path, step, step location, foot plant, foothold, future foothold, foot location, future foot location, foot position. I think some terms are being used interchangeably. I would really highly recommend a diagrammatic cartoon sketch, showing the definitions of all these terms in a single figure, and then sticking to them in the main text.

We have made the language more systematic and clarified the definition of each term (see Methods). Path refers to the sequence of 5 steps. Foothold is where the foot was placed in the environment. A step is the transition from one foothold to the next.

(3) More coverage of different interpretations / less interpretation in the abstract/introduction would be prudent. The authors discuss the path selection very much on the basis of energetic costs and gait stability. At least mention should be given to other plausible parameters the participants might be optimizing (or that indeed they may be just satisficing). That is, it is taken as "given" that energetic cost is the major driver of path selection in your task, and that the relevant perception relies on internal models. Neither of these is a priori obvious nor is it as far as I can tell shown by the data (optimizing other variables, satisficing behavior, or online "direct perception" cannot be ruled out).

The abstract has been substantially rewritten. We have adjusted our language in the introduction/discussion to try to address this concern.

**Recommendations for the authors:**

**Reviewing Editor comments**
You will find a full summary of all 3 reviews below. In addition to these reviews, I'd like to highlight a few points from the discussion among reviewers.All reviewers are in agreement that this study has the potential to be a fundamental study with far-reaching empirical and practical implications. The reviewers also appreciate the technical achievements of this study.At the same time, all reviewers are concerned with the overall lack of clarity in how the results are presented. There are a considerable number of figures that need better labeling, text parts that require clearer definitions, and the description of data collection and analysis (esp. with regard to the CNN) requires more care. Please pay close attention to all comments related to this, as this was the main concern that all reviewers shared.At a more specific level, the reviewers discussed the finding around leg length, and admittedly, found it hard to believe, in short: "extraordinary claims need strong evidence". It would be important to strengthen this analysis by considering possible confounds, and by including a discussion of the degree of conviction.

We have weakened the discussion of this finding and provided some an additional analyses in a supplemental figure (Figure 17) to help clarify the finding.

**Reviewer #1 (Recommendations For The Authors):**
First, let me apologize for the long delay with this review. Despite my generally positive evaluation (see public review), I have some concerns about the way the data are presented and questions about methodological details.(1) Representation of results: I find it hard to decipher how much variability arises within an individual and how much across individuals. For example, Figure 7b seems to aggregate across all individuals, while the analysis is (correctly) based on the subject medians.

Figure 7b That figure was just one subject. This is now clarified.

It would be good to see the distribution of all individuals (maybe use violin plots for each observer with the true data on one side and the baseline data on the other, or simple histograms for each). To get a feeling for inter-individual and intra-individual variability is crucial, as obviously (see the leg-length analysis) there are larger inter-individual differences and representations like these would be important to appreciate whether there is just a scaling of more or less the same effect or whether there are qualitative differences (especially in the light of N=9 being not a terribly huge sample size).

The medians for the individual subjects are now provided with the standard deviations between subjects to indicate the extent of individual differences. Note that the random paths were chosen from the distribution of actual step slopes for that subject as one of the constraints. This makes the random paths statistically similar to the chosen paths with the differences only being generated by the particular visual context. Thus the test for a difference between chosen and random is quite conservative

Similarly, seeing \DeltaH plotted as a function of steps in the path as a figure rather than just having the verbal description would also help.

To simplify the discussion of our methods/results we have removed the analyses that examine mean slope as a function of steps. Because of the central limit theorem the slopes of the chosen paths remain largely unchanged regardless of the choice path length. The slopes of the simulated paths are always larger irrespective of the choice of path length.

(2) Reporting the statistical analyses: This is related to my previous issue: I would appreciate it if the test statistics and degrees-of-freedom of the statistical tests were given along with the p-values, instead of only the p-values. This at some points would also clarify how the statistics were computed exactly (e.g., "All subjects showed comparable difference and the difference in medians evaluated across subjects was highly significant (p<<0.0001).", p.10, is ambiguous to me).

Details have been added as requested.

(3) Why is the lower half ("tortuosity less than the median tortuosity" of paths used as "straight" rather than simply the minimum of all viable paths)?

The benchmark for a straight path is somewhat arbitrary. Using the lower half rather than the minimum length path is more conservative.

(4) For the CNN analysis, I failed to understand what was training and what was test set. I understand that the goal is to predict for all pixels whether they are a potential foothold or not, and the AUC is a measure of how well they can be discriminated based on depth information and then this is done for each image and the median over all images taken. But on which data is the CNN trained, and on which is it tested? Is this leave-n-out within the same participant? If so, how do you deal with dependencies between subsequent images? Or is it leave-1-out across participants? If so, this would be more convincing, but again, the same image might appear in training and test. If the authors just want to ask how well depth features can discriminate footholds from non-footholds, I do not see the benefit of a supervised method, which leaves the details of the feature combinations inside a black box. Rather than defining the "negative set" (i.e., the non-foothold pixels) randomly, the simulated paths could also be used, instead. If performance (AUC) gets lower than for random pixels, this would confirm that the choice of parameters to define a "viable path" is well-chosen.

This has been clarified as described above.

Minor issues:(5) A higher tortuosity would also lead a participant to require more steps in total than a lower tortuosity. Could this partly explain the correlation between the leg length and the slope/tortuosity correlation? (Longer legs need fewer steps in total, thus there might be less tradeoff between \Delta H and keeping the path straight (i.e., saving steps)). To assess this, you could give the total number of steps per (straight) distance covered for leg length and compare this to a flat surface.

The calculations are done on an individual subject basis and the first and last step locations are chosen from the actual foot placements, then the random paths are generated between those endpoints. The consequence of this is that the number of steps is held constant for the analysis. We have clarified the methods for this analysis to try to make this more clear.

(6) As far as I understand, steps happen alternatingly with the two feet. That is, even on a flat surface, one would not reach 0 tortuosity. In other words, does the lateral displacement of the feet play a role (in particular, if paths with even and paths with odd number of steps were to be compared), and if so, is it negligible for the leg-length correlation?

All the comparisons here are done for 5 step sequences so this potential issue should not affect the slope of the regression lines or the leg length correlation.

(7) Is there any way to quantify the quality of the depth estimates? Maybe by taking an actual depth image (e.g., by LIDAR or similar) for a small portion of the terrain and comparing the results to the estimate? If this has been done for similar terrain, can a quantification be given? If errors would be similar to human errors, this would also be interesting for the interpretation of the visual sampling data.

Unfortunately, we do not have the ground truth depth image from LIDAR. When these data were originally collected, we had not imagined being able to reconstruct the terrain. However, we agree with the reviewers that this would be a good analysis to do. We plan to collect LIDAR in future experiments.

To provide an assessment of quality for these data in the absence of a ground truth depth image, we have performed an evaluation of the reliability of the terrain reconstruction across repeats of the same terrain both between and within participants. We have expanded the discussion of these reliability analyses in the results section entitled “Evaluating Terrain Reconstruction”, as well as in the corresponding methods section (see Figure 10).

(8) The figures are sometimes confusing and a bit sloppy. For example, in Figure 7a, the red, cyan, and green paths are not mentioned in the caption, in Figure 8 units on the axes would be helpful, in Figure 9 it should probably be "tortuosity" where it now states "curviness".

These details have been fixed.

(9) I think the statement "The maximum median AUC of 0.79 indicates that the 0.79 is the median proportion of pixels in the circular..." is not an appropriate characterization of the AUC, as the number of correctly classified pixels will not only depend on the ROC (and thus the AUC), but also on the operating point chosen on the ROC (which is not specified by the AUC alone). I would avoid any complications at this point and just characterize the AUC as a measure of discriminability between footholds and non-footholds based on depth features.

This has been fixed.

(10) Ref. [16]is probably the wrong Hart paper (I assume their 2012 Exp. Brain Res. [https://doi.org/10.1007/s00221-012-3254-x] paper is meant at this point)

Fixed

Typos (not checked systematically, just incidental discoveries):(11) "While there substantial overlap" (p.10)(12) "field.." (p.25)(13) "Introduction", "General Discussion" and "Methods" as well as some subheadings are numbered, while the other headings (e.g., Results) are not.

Fixed

**Reviewer #2 (Recommendations For The Authors):**
The major suggestions have been made in the Public Review. The following are either minor comments or go into more detail about the major suggestions. All of these comments are meant to be constructive, not obstructive.Abstract. This is well written, but the main conclusions "Walkers avoid...This trade off is related...5 steps ahead" sound quite qualitative. They could be strengthened by more specificity (NOT p-values), e.g. "positive correlation between the unevenness of the path straight ahead and the probability that people turned off that path."

The abstract has been substantially rewritten.

P. 5 "pinning the head position estimated from the IMU to the Meshroom estimates" sounds like there are two estimates. But it does not sound like both were used. Clarify, e.g. the Meshroom estimate of head position was used in place of IMU?

Yes that’s correct. We have clarified this in the text.

Figure 5. I was confused by this. First, is a person walking left to right? When the gaze position is shown, where was the eye at the time of that gaze? There are straight lines attached to the blue dots, what do they represent? The caption says gaze is directed further along the path, which made me guess the person is walking right to left, and the line originates at the eye. Except the origins do not lie on or close to the head locations. There's also no scale shown, so maybe I am completely misinterpreting. If the eye locations were connected to gaze locations, it would help to support the finding that people look five steps ahead of where they step.

We have updated the figure and clarified the caption to remove these confusions. There was a mistake in the original figure (where the yellow indicated head locations, we had plotted the center of mass and the choice of projection gave the incorrect impression that the fixations off the path, in blue, were separated from the head).

The view of the data is now presented so the person is walking left to right and with a projection of the head location (orange), gaze locations (blue or green) and feet (pink).

Figure 6. As stated in the major comments, the step distributions would be expected to have a covariance structure (in terms of raw data before taking absolute values). It would be helpful to report the covariances (6 numbers). As an example of a simple statistical analysis, a PCA (also based on a data covariance) would show how certain combinations of slope/distance/direction are favored over others. Such information would be a simple way to argue that the data are not completely random, and may even show a height-turn trade-off immediately. (By the way, I am assuming absolute values are used because the slopes and directions are only positive, but it wasn't clear if this was the definition). A reason why covariances and PCA are helpful is that such data would be helpful to compute a better random walk, generated from dynamics. I believe the argument that steps are not random is not served by showing the different histograms in Figure 7, because I feel the random paths are not fairly produced. A better argument might draw randomly from the same distribution as the data (or drive a dynamical random walk), and compare with actual data. There may be correlations present in the actual data that differ from random. I could be mistaken, because it is difficult or impossible to draw conclusions from distributions of absolute values, or maybe I am only confused. In any case, I suspect other readers will also have difficulty with this section.

This has been addressed above in the major comments.

p. 9, "average step slope" I think I understand the definition, but I suggest a diagram might be helpful to illustrate this.

There is a diagram of a single step slope in Figure 6 and a diagram of the average step slope for a path segment in Figure 12.

Incidentally, the "straight path slope" is not clearly defined. I suspect "straight" is the view from above, i.e. ignoring height changes.

Clarified

p. 11 The tortuosity metric could use a clearer definition. Should I interpret "length of the chosen path relative to a straight path" as the numerator and denominator? Here does "length" also refer to the view from above? Why is tortuosity defined differently from step slope? Couldn't there be an analogue to step slope, except summing absolute values of direction changes? Or an analogue to tortuosity, meaning the length as viewed from the side, divided by the length of the straight path?

We followed the literature in the definition of tortuosity. We have clarified the definition of tortuosity in the methods, but yes, you can interpret the length of the chosen path relative to a straight path, as the numerator and denominator, and length refers to 3D length. We agree that there are many interesting ways to look at the data but for clarity we have limited the discussion to a single definition of tortuosity in this paper.

Figure 8 could use better labeling. On the left, there is a straight path and a more tortuous path, why not report the metrics for these? On the right, there are nine unlabeled plots. The caption says "turn probability vs. straight path slope" but the vertical axis is clearly not a probability. Perhaps the axis is tortuosity? I presume the horizontal axis is a straight path slope in degrees, but this is not explained. Why are there nine plots, is each one a subject? I would prefer to be informed directly instead of guessing. (As a side note, I like the correlations as a function of leg length, it is interesting, even if slightly unbelievable. I go hiking with people quite a bit shorter and quite a lot taller than me, and anecdotally I don't think they differ so much from each other).

We have fixed Figure 8 which shows the average “mean slope” as a function of tortuosity. We have added a supplemental figure which shows a scatter plot of the raw data (mean slope vs. tortuosity for each path segment).

Note that when walking with friends other factors (e.g. social) will contribute to the cost function. As a very short person my experience is that it is a problem. In any case, the data are the data, whatever the underlying reasons. It does not seem so surprising that people of different heights make different tradeoffs. We know that the preferred gait depends on individual’s passive dynamics as described in the paper, and the terrain will change what is energetically optimal as described in the Darici and Kuo paper.

Figure 9 presumably shows one data point per subject, but this isn't clear.

The correlations are reported per subject, and this has been clarified.

p. 13 CNN. I like this analysis, but only sort of. It is convincing that there is SOME sort of systematic decision-making about footholds, better than chance. What it lacks is insight. I wonder what drives peoples' decisions. As an idle suggestion, the AlexNet (arXiv: Krizhevsky et al.; see also A. Karpathy's ConvNETJS demo with CIFAR-10) showed some convolutional kernels to give an idea of what the layers learned.

Further exploration of CNN’s would definitely be interesting, but it is outside the scope of the paper. We use it simply to make a modest point, as described above.

p. 15 What is the definition of stability cost? I understand energy cost, but it is unclear how circuitous paths have a higher stability cost. One possible definition is an energetic cost having to do with going around and turning. But if not an energy cost, what is it?

We meant to say that the longer and flatter paths are presumably more stable because of the smaller height changes. You are correct that we can’t say what the stability cost is and we have clarified this in the discussion.

p. 16 "in other data" is not explained or referenced.

Deleted

p. 10 5 step paths and p. 17 "over the next 5 steps". I feel there is very little information to really support the 5 steps. A p-value only states the significance, not the amount of difference. This could be strengthened by plotting some measures vs. the number of steps ahead. For example, does a CNN looking 1-5 steps ahead predict better than one looking N<5 steps ahead? I am of course inclined to believe the 5 steps, but I do not see/understand strong quantitative evidence here.

We have weakened the statements about evidence for planning 5 steps ahead.

p. 25 CNN. I did not understand the CNN. The list of layers seems incomplete, it only shows four layers. The convolutional-deconvolutional architecture is mentioned as if that is a common term, which I am unfamiliar with but choose to interpret as akin to encoder-decoder. However, the architecture does not seem to have much of a bottleneck (25x25x8 is not greatly smaller than 100x100x4), so what is the driving principle? It's also unclear how the decoder culminates, does it produce some m x m array of probabilities of stepping, where m is some lower dimension than the images? It might be helpful also to illustrate the predictions, for example, show a photo of the terrain view, along with a probability map for that view. I would expect that the reader can immediately say yes, I would likely step THERE but not there.

We have clarified the description of the CNN. An illustration is shown in Figure 11.

**Reviewer #3 (Recommendations For The Authors):**
(This section expands on the points already contained in the Public Review).Major issues(1) The training and validation data of the CNN are not explained fully making it unclear if the data tells us anything about the visual features used to guide steering. A CNN was used on the depth scenes to identify foothold locations in the images. This is the bit of the methods and the results that remains ambiguous, and the authors may need to revisit the methods/results. It is not clear how or on what data the network was trained (training vs. validation vs. un-peeked test data), and justification of the choices made. There is no discussion of possible overfitting. The network could be learning just for example specific rock arrangements in the particular place you experimented. Training the network on data from one location and then making it generalize to another location would of course be ideal. Your network probably cannot do this (as far as I can tell this was not tried), and so the meaning of the CNN results cannot really be interpreted.I really like the idea, of getting actual retinotopic depth field approximations. But then the question would be: what features in this information are relevant and useful for visual guidance (of foot placement)? But this question is not answered by your method."If a CNN can predict these locations above chance using depth information, this would indicate that depth features can be used to explain some variation in foothold selection." But there is no analysis of what features they are. If the network is overfitting they could be very artefactual, pixel-level patterns and not the kinds of "features" the human reader immediately has in mind. As you say "CNN analysis shows that subject perspective depth features are predictive of foothold locations", well, yes, with 50,000 odd parameters the foothold coordinates can be associated with the 3D pixel maps, but what does this tell us?

See previous discussion of these issues.

It is true that we do not know the precise depth features used. We established that information about height changes was being used, but further work is needed to specify how the visual system does this. This is mentioned in the Discussion.

You open the introduction with a motivation to understand the visual features guiding path selection, but what features the CNN finds/uses or indeed what features are there is not much discussed. You would need to bolster this, or down-emphasize this aspect in the Introduction if you cannot address it."These depth image features may or may not overlap with the step slope features shown to be predictive in the previous analysis, although this analysis better approximates how subjects might use such information." I do not think you can say this. It may be better to approximate the kind of (egocentric) environment the subjects have available, but as it is I do not see how you can say anything about how the subject uses it. (The results on the path selection with respect to the terrain features, viewpoint viewpoint-independent allocentric properties of the previous analyses, are enough in themselves!)

We have rewritten the section on the CNN to make clearer what it can and cannot do and its role in the manuscript. See previous discussion.

(2) The use of descriptive terminology should be made systematic. Overall the rest of the methodology is well explained, and the workflow is impressive. However, to interpret the results the introduction and discussion seem to use terminology somewhat inconsistently. You need to dig into the methods to figure out the exact operationalizations, and even then you cannot be quite sure what a particular term refers to. Specifically, you use the following terms without giving a single, clear definition for them (my interpretation in parentheses):foothold (a possible foot plant location where there is an "affordance"? or a foot plant location you actually observe for this individual? or in the sample?)step (foot trajectory between successive step locations)step location (the location where the feet are placed)path are they lines projected on the ground, or are they sequences of foot plants? The figure suggests lines but you define a path in terms of five steps.foot plant (occurs when the foot comes in contact with step location?)future foothold (?)foot location (?)future foot location (?)foot position (?)I think some terms are being used interchangeably here? I would really highly recommend a diagrammatic cartoon sketch, showing the definitions of all these terms in a single figure, and then sticking to them in the main text. Also, are "gaze location" and "fixation" the same? I.e. is every gaze-ground intersection a "gaze location" (I take it it is not a "fixation", which you define by event identification by speed and acceleration thresholds in the methods)?

We have cleaned up the language. A foothold is the location in the terrain representation (mesh) where the foot was placed. A step is the transition from one foothold to the next. A path is the sequences of 5 steps. The lines simply illustrate the path in the Figures. A gaze location is the location in the terrain representation where the walker is holding gaze still (the act of fixating). See Muller et al (2023) for further explanation.

(3) More coverage of different interpretations / less interpretation in the abstract/introduction would be prudent. You discuss the path selection very much on the basis of energetic costs and gait stability. At least mention should be given to other plausible parameters the participants might be optimizing (or that indeed they may be just satisficing). Temporal cost (more circuitous route takes longer) and uncertainty (the more step locations you sample the more chance that some of them will not be stable) seem equally reasonable, given the task ecology / the type of environment you are considering. I do not know if there is literature on these in the gait-scene, but even if not then saying you are focusing on just one explanation because that's where there is literature to fall back on would be the thing to do.Also in the abstract and introduction you seem to take some of this "for granted". E.g. you end the abstract saying "are planning routes as well as particular footplants. Such planning ahead allows the minimization of energetic costs. Thus locomotor behavior in natural environments is controlled by decision mechanisms that optimize for multiple factors in the context of well-calibrated sensory and motor internal models". This is too speculative to be in the abstract, in my opinion. That is, you take as "given" that energetic cost is the major driver of path selection in your task, and that the relevant perception relies on internal models. Neither of these is a priori obvious nor is it as far as I can tell shown by your data (optimizing other variables, satisficing behavior, or online "direct perception" cannot be ruled out).

We have rewritten the abstract and Discussion with these concerns in mind.

You should probably also reference:Warren, W. H. (1984). Perceiving affordances: Visual guidance of stair climbing. Journal of Experimental Psychology: Human Perception and Performance, 10(5), 683-703. https://doi.org/10.1037/0096-1523.10.5.683Warren WH Jr, Young DS, Lee DN. Visual control of step length during running over irregular terrain. J Exp Psychol Hum Percept Perform. 1986 Aug;12(3):259-66. doi: 10.1037//0096-1523.12.3.259. PMID: 2943854.

We have added these references to the introduction.

Minor pointRelated to (2) above, the path selection results are sometimes expressed a bit convolutedly, and the gist can get lost in the technical vocabulary. The generation of alternative "paths" and comparison of their slope and tortuousness parameters show that the participants preferred smaller slope/shorter paths. So, as far as I can tell, what this says is that in rugged terrain people like paths that are as "flat" as possible. This is common sense so hardly surprising. Do not be afraid to say so, and to express the result in plain non-technical terms. That an apple falls from a tree is common sense and hardly surprising. Yet quantifying the phenomenon, and carefully assessing the parameters of the path that the apple takes, turned out to be scientifically valuable - even if the observation itself lacked "novelty".

Thanks. We have tried to clarify the methods/results with this in mind.